

# An evaluation of European nitrogen and sulfur wet deposition and their trends estimated by six chemistry transport models for the period 1990–2010

Mark R. Theobald[1], Marta G. Vivanco[1], Wenche Aas[2], Camilla Andersson[3], Giancarlo Ciarelli[4], Florian Couvidat[5], Kees Cuvelier[6], Astrid Manders[7], Mihaela Mircea[8], Maria-Teresa Pay[9], Svetlana Tsyro[10], Mario Adani[8], Robert Bergström[3,11], Bertrand Bessagnet[5], Gino Briganti[8], Andrea Cappelletti[8], Massimo D'Isidoro[8], Hilde Fagerli[10], Kathleen Mar[12], Noelia Otero[12], Valentin Raffort[13], Yelva Roustan[13], Martijn Schaap[7,14], Peter Wind[10,15] and Augustin Colette[5]

[1]Atmospheric Pollution Unit, CIEMAT, Avda. Complutense, 40, 28040 Madrid, Spain
[2]Norwegian Institute for Air Research (NILU), Box 100, 2027 Kjeller, Norway
[3]Swedish Meteorological and Hydrological Institute, 60176 Norrköping, Sweden
[4]Laboratoire Inter-Universitaire des Systèmes Atmosphériques (LISA), UMR CNRS 7583, Université Paris Est Créteil et Université Paris Diderot, Institut Pierre Simon Laplace, Créteil, France
[5]National Institute for Industrial Environment and Risks (INERIS), Parc Technologique ALATA, F-60550 Verneuil-en-Halatte, France
[6]ex European Commission, Joint Research Centre (JRC), Ispra, Italy
[7]Netherlands Organisation for applied scientific research (TNO), P.O. Box 80015, 3508 TA Utrecht, The Netherlands
[8]Italian National Agency for New Technologies, Energy and Sustainable Economic Development (ENEA), Via Martiri di Monte Sole 4, 40129 Bologna, Italy
[9]Barcelona Supercomputing Center, Centro Nacional de Supercomputación, Jordi Girona, 29, 08034 Barcelona, Spain
[10]Climate Modelling and Air Pollution Division, Research and Development Department, Norwegian Meteorological Institute (MET Norway), Blindern, N-0313 Oslo, Norway
[11]Chalmers University of Technology, Gothenburg, SE-412 96, Sweden
[12]Institute for Advanced Sustainability Studies (IASS), Postdam, Germany
[13]CEREA, Joint Laboratory Ecole des Ponts ParisTech - EDF R&D, Champs-Sur-Marne, France
[14]Institute for Meteorology, Free University Berlin, Berlin, Germany
[15]Faculty of Science and Technology, University of Tromsø, Tromsø, Norway

*Correspondence to*: Mark R. Theobald (mark.theobald@ciemat.es)

**Abstract.** The wet deposition of nitrogen and sulfur in Europe for the period 1990–2010 was estimated by six atmospheric chemistry transport models (CHIMERE, CMAQ, EMEP MSC-W, LOTOS-EUROS, MATCH and MINNI) within the framework of the EURODELTA-Trends model intercomparison. The simulated wet deposition and its trends for two eleven-year periods (1990–2000 and 2000–2010) were evaluated using data from observations from the EMEP European monitoring network. For annual wet deposition of oxidised nitrogen (WNOx), model bias was within 30% of the average of the observations for most models. There was a tendency for most models to underestimate annual wet deposition of reduced nitrogen (WNHx) although model bias was within 40% of the average of the observations. Model bias for WNHx was inversely correlated with model bias for atmospheric concentrations of $NH_3 + NH_4^+$, suggesting that an underestimation of





wet deposition partially contributed to an overestimation of atmospheric concentrations. Model bias was also within about 40% of the average of the observations for the annual wet deposition of sulfur (WSOx) for most models.

Decreasing trends in WNOx were observed at most sites for both eleven-year periods, with larger trends, on average, for the second period. The models also estimated predominantly decreasing trends at the monitoring sites and all but one of the models estimated larger trends, on average, for the second period. Decreasing trends were also observed at most sites for WNHx, although larger trends, on average, were observed for the first period. This pattern was not reproduced by the models, which estimated smaller decreasing trends, on average, than those observed or even small increasing trends. The largest observed trends were for WSOx, with decreasing trends at more than 80% of the sites. On average, the observed trends were larger for the first period. All models were able to reproduce this pattern although some models underestimated the trends (by up to a factor of four) and others overestimated them (by up to 40%), on average. These biases in modelled trends were directly related to the tendency of the models to under- or overestimate annual wet deposition and were smaller for the relative trends (expressed as % yr$^{-1}$ relative to the deposition at the start of the period).

The fact that model biases were fairly constant throughout the time series makes it possible to improve the predictions of wet deposition for future scenarios by adjusting the model estimates using a bias correction calculated from past observations. An analysis of the contributions of various factors to the modelled trends suggests that the predominantly decreasing trends in wet deposition are mostly due to reductions in emissions of the precursors $NO_x$, $NH_3$ and $SO_x$. However, changes in meteorology (e.g. precipitation) and other (non-linear) interactions partially offset the decreasing trends due to emission reductions during the first period, but not the second. This suggests that the emission reduction measures had a larger effect on wet deposition during the second period, at least for the sites with observations.

## 1 Introduction

Atmospheric deposition of nitrogen (N) and sulfur (S) can lead to the acidification of soils and surface waterways resulting in damage to semi-natural vegetation and aquatic organisms (Ulrich, 1983). Nitrogen deposition can also lead to the eutrophication of terrestrial and aquatic ecosystems, resulting in a reduction in biodiversity (Bobbink et al., 1998). Most of the deposited N and S originates from the emissions of nitrogen oxides ($NO_x$), ammonia ($NH_3$) and sulfur dioxide ($SO_2$), which through chemical reactions, can form aerosol species, such as ammonium nitrate ($NH_4NO_3$) and ammonium sulfate (($NH_4$)$_2SO_4$) during atmospheric transport. The resulting gaseous and particulate nitrogen- and sulfur-containing compounds in the atmosphere can be subsequently deposited to the surface through the mechanisms of wet (in rain, fog or snow) and dry deposition. Within Europe, most (60–95%) of the N and S deposition is estimated to come from European emissions (Sanderson et al., 2008; Tan et al., 2018; Vivanco et al., 2018). Over the last three decades there has been a reported decrease in European emissions of $NO_x$, $NH_3$ and $SO_2$ by approximately 50%, 15% and 90%, respectively (EEA, 2017). These changes have mostly occurred through the implementation of control measures under the UNECE Convention on Long-



range Transboundary Air Pollution (UNECE, 1979) and the European Union National Emission Ceilings Directive (Directive 2001/81/EC). Over this time period, decreases in nitrogen and sulfur deposition would be expected in response to the emission reductions. Indeed, an analysis of data from the EMEP monitoring network revealed significant decreases in non-sea-salt sulphate concentrations in precipitation at more than 70% of sites for the periods 1990–2001 and 2002–2012 (Colette et al., 2016). Over the two periods the median decrease in concentrations was approximately 50%. Decreasing trends of oxidised and reduced inorganic N in precipitation were also observed for the two periods but the trends at the majority of the sites were not significant. Oxidised nitrogen decreased by an average of 19% and 23% and reduced nitrogen decreased by an average of 30% and 16%, for the two periods, respectively. Tørseth et al. (2012) also analysed the EMEP data for the period 1990–2009 and found mean decreasing trends of non-sea-salt sulfate and oxidised and reduced nitrogen in precipitation of 64%, 25% and 25% respectively. At the beginning of this period EMEP sites were mostly located in central Europe, UK and Scandinavian and so the calculated trends may not be representative of southern and eastern parts of Europe.

Atmospheric chemistry transport models (CTMs) can be used to study the relationships between emissions of $NO_x$, $NH_3$ and $SO_2$ and the wet and dry deposition of N and S, since they simulate the main processes influencing the fate of atmospheric pollutants (turbulent dispersion, atmospheric chemistry, cloud processes, long-range transport, wet and dry deposition, etc.). Although they are no substitute for observations, CTM simulations have the advantage of estimating deposition rates for locations where there are no measurements and for processes for which measurements are difficult and/or sparse (e.g. dry deposition). They can also be used for simulating hypothetical scenarios, such as the effect of emission reduction strategies. However, in order to provide reliable estimates for such scenarios, the deposition estimated by CTMs needs to be evaluated for real situations with existing measurement data. In Europe, this evaluation can currently only be done for wet deposition since measurements of dry deposition of N and S are sparse, incomplete and intermittent. Along these lines, Simpson et al. (2006) compared N and S wet deposition estimated by the EMEP MSC-W CTM with measurement data from 160 sites of the European Union/ICP Forest Level II monitoring network for the years 1997 and 2000. They found that the model underestimated deposition rates, on average, although the mean model estimates were within 20–30% of the mean observed deposition. Similarly, Simpson et al. (2014) compared the wet deposition estimates of three European-scale CTMs (EMEP MSC-W, MATCH and SILAM) and one hemispheric CTM (DEHM) driven by climate-model-based meteorology with observations from more than 80 sites of the EMEP network for the period 2000–2010 and found that these models generally estimated mean wet deposition rates of oxidised N, reduced N and S within 30% of the mean observed values. Exceptions were overestimations of S deposition by EMEP and MATCH (by 35% and 82%, respectively) and an overestimation of oxidised N deposition by SILAM of 72%.

More recently, Vivanco et al. (2017) compared the wet deposition estimates of six CTMs with observations made at more than 40 sites in the EMEP network during four one-month campaigns. A large variability was found between model





performance for the different components of wet deposition. For reduced N, all models underestimated the mean wet deposition, although by varying degrees (2–62%). By contrast, some models overestimated the mean wet deposition of oxidised N by up to 11%, while others underestimated it by more than 40%. Model performance for S wet deposition was even more variable with mean model biases ranging from -40% to +50%. Another study comparing the wet deposition estimates from the EMEP network with the estimates made by 13 CTMs for 2010 (including the simulations used in the present study) found that most of the models underestimated wet deposition, with negative mean model biases of up to 70%, 86% and 58% for S, oxidised N and reduced N (Vivanco et al., 2018).

Dentener et al. (2006) compared the ensemble mean wet deposition of 23 global CTMs with observations at more than 40 EMEP sites for the year 2000. The authors found that the ensemble underestimated European wet deposition of oxidised N and S (by 9–10%), on average, but overestimated that of reduced N by 6%. However, more recently, Vet et al. (2014) found that an ensemble of 21 global CTMs underestimated wet deposition of oxidised and reduced N by about 20%, on average, and overestimated that of S by about 10% when compared with observations at more than 100 EMEP sites for the period 2000–2002.

The variability in model performance for wet deposition is not surprising since wet deposition depends on many processes, such as emissions, dispersion, atmospheric chemistry, cloud formation, cloud chemistry and precipitation, etc. However, despite their inherent uncertainties, CTMs are useful tools to complement observations and investigate the spatial distributions of atmospheric deposition and their evolution over time. One key question is how well the models can simulate the trends in deposition as a result of changes in emissions. This aspect is important since CTMs are frequently used to evaluate the impact of future emission control measures and so model estimates of future deposition rates need to be reliable in order to make well-founded policy decisions. In addition to the EMEP trends report (Colette et al., 2016), several studies have looked at trends in observed deposition (or precipitation chemistry) in Europe. For example, Waldner et al. (2014) found significant decreasing trends in bulk deposition of sulfate at about 80% of ICP Forests Level II sites for the period 2001–2010, with a mean annual trend of -4% relative to the 2001 deposition. By contrast, the authors only found significant trends in oxidised and reduced N deposition at 37% and 26% of the sites, with mean annual trends of -1.4% and -0.9%, respectively. Similarly, Pascaud et al. (2016) found significant decreasing trends in non-sea-salt sulfate deposition over the period 1995–2007 at about 70% of measurement sites in France, with a mean annual decrease in deposition of 3.3%. They also found fewer sites with significant trends of oxidised and reduced N deposition, with significant decreasing trends at 24% and 11% of the sites, respectively. Marchetto et al. (2013) also found significant decreasing trends in precipitation sulfate at eight out of nine sites in Italy for the period 1998–2010. However, in contrast to the studies cited above, they also found significant decreasing trends for nitrate and ammonium at most of the sites. This could to be due to the fact that they calculated the trends from weekly data and not monthly or seasonal values. Fagerli and Aas (2008) used annual data to calculate the trends of ammonium and nitrate in precipitation for the period 1980–2003 at 24 sites of the EMEP network.



The authors found significant decreasing trends of precipitation nitrate and ammonium at more than 70% of the sites, with mean annual trends of -1.3% and -1.9%, respectively. They also compared the observed trends with those calculated from simulations by the EMEP Unified model for the years for which the model was run (1980, 1985, 1990 and 1995–2003). Comparing the model results with the observations for the these years showed that significant decreasing modelled trends for

nitrate and ammonium in precipitation were estimated for 54% and 50% of the sites, whereas significant observed trends were only found at 33% of the sites for both compounds. However, modelled and observed trends in precipitation nitrate averaged over all sites were similar (-1.4% vs. -1.6% yr$^{-1}$) but modelled trends in precipitation ammonium were, on average, smaller than those observed (-1.2% vs. -2.1% yr$^{-1}$). Enghardt et al. (2017) also compared modelled (EMEP MSC-W and MATCH) and observed concentrations of oxidised and reduced N and of S in precipitation for the period 1955–2010 for

sites in the EMEP network (and its predecessor the European Air Chemistry Network). They found that the models estimate the relative trends fairly well since the mid-1980s. These last two studies appear to be the only ones that have compared modelled and observed trends in wet-deposition in Europe.

The EURODELTA-Trends (EDT) exercise aims to assess the role of European air pollutant emission reductions in improving air quality and reducing  the acidification and eutrophication of ecosystems over the past two decades (Colette et

al., 2017a), as well as assess the influence of meteorological variability and long range transport through the boundary conditions used. Eight CTMs were used to simulate air quality over the period 1990–2010, of which six delivered estimates of wet and dry deposition of N and S, thus providing a unique dataset for testing the ability of multiple CTMs to simulate deposition trends.

In this paper, we compare the EDT CTMs' estimates of wet deposition of S and reduced and oxidised N with observations

made within the EMEP network over the period 1990–2010. In order to better understand the differences between the CTMs' estimates of wet deposition, we also evaluate the models for atmospheric concentrations of relevant gaseous and particulate species and seasonal precipitation rates, as well as compare the model estimates for dry deposition. Due to the number of models studied and the many differences between their formulations and parameterisations, it is out of the scope of this study to provide an in depth analysis of individual model performance or inter-model differences. We also evaluate

the ability of the models to estimate the absolute and relative trends in wet deposition over two eleven-year periods (1990–2000 and 2000–2010) and look at the contributions that changing emissions, boundary conditions and meteorology make to the overall modelled trends. Following a discussion of uncertainties and limitations associated with the model simulations and the observations of wet deposition, we provide suggestions for how to improve model estimates of wet deposition in the future.



## 2 Methods

### 2.1 Model simulations

Six CTMs were used to estimate wet deposition in Europe for the period 1990–2010: Chimere (Couvidat et al., 2018), CMAQ (Byun and Schere, 2006), EMEP MSC-W (Simpson et al., 2012), LOTOS-EUROS (Manders et al., 2017), MATCH (Robertson et al., 1999) and MINNI (Mircea et al., 2014, 2016). The shortened model names CHIM, CMAQ, EMEP, LOTO, MATCH and MINNI, respectively, are used throughout this paper. An overview of the model chemistry schemes and parameterisations for wet and dry deposition can be found in the supplementary material (Table S1). In order to assess the differences in model estimates due only to model structure and parameterisations, the modelling domain and input data used in the simulations were the same for all models, wherever possible. The models were run on a domain that covers most of Europe (Fig. 1) with a grid resolution of 0.25° in latitude and 0.4° in longitude with the exception of CMAQ, which used a Lambert Conformal Conic Projection with a grid resolution of 25 km × 25 km. Most of the CTMs used the same meteorological data from hindcast simulations related to the EuroCordex Project (Jacob et al., 2014; Stegehuis et al., 2015) by the Weather Research and Forecast (WRF) model (Skamarock et al., 2008) at a spatial resolution of 0.44° nudged towards the ERA-Interim reanalyses (Dee et al., 2011). The exceptions were CMAQ, which used data from WRF with an identical set-up on a Lambert Conformal projection and LOTO and MATCH, which used the ERA-Interim reanalyses further downscaled with RACMO2 (van Meijgaard, 2012) and HIRLAM (Dahlgren et al., 2016), respectively. The latter also included a reanalysis, forced to the ERA-Interim reanalyses at the boundaries. All models used the same gridded anthropogenic emissions. These were derived from national annual emissions for $SO_2$, $NOx$, non-methane volatile organic compounds, CO, $NH_3$ and particulate matter ($PM_{2.5}$, $PM_{10}$, black carbon and organic carbon) estimated by the Greenhouse gases and Air pollution INteractions and Synergies (GAINS) model (Amann et al., 2011). This scenario (ECLIPSE_V5) is freely available from the webpage of the online version of the GAINS model: http://gains.iiasa.ac.at/models/. Emission data are available for the years 1990, 1995, 2000, 2005 and 2010 and the intermediate years were calculated by linear interpolation. The national emissions were spatially disaggregated to the EDT grid using proxies for roads and shipping routes and population density, the European Pollutant Release and Transfer Register (EPRTR) and the TNO-MACC inventory for $NH_3$ emissions (Terrenoire et al., 2015; Bessagnet et al., 2016). Where high spatial resolution inventories are available (UK and France), the national data were used to disaggregate the emissions. During the post-processing of the simulation output, an error was detected in the emissions of primary particulate matter for Russia, North Africa and maritime areas for the period 1991-1999. Since this error was identified late in the analysis process it was not possible to re-run the simulations with corrected emissions. However, an analysis of the impact of this error on modelled wet deposition was carried out using the CHIMERE model (see Sect. S1 of the supplementary material for more details). From this analysis we estimate that the errors in wet deposition due to the errors in emissions are less than 0.5% for most of the modelling domain with maximum errors of less than 2.5%. These errors are small compared with the overall uncertainty of the model estimates



and the uncertainty of the observations. Biogenic and natural emissions were not prescribed and each model used their own set-up. The boundary conditions used in the models were a simplified version of those used in the standard EMEP MSC-W model based on a climatology of observational data (Simpson et al., 2012). Full details of the models, input data and boundary conditions used can be found in Colette et al. (2017a). One model (CMAQ) only simulated the years 1990, 2000

and 2010, whereas the other models simulated all 21 years (1990–2010). For the evaluation of the model estimates of wet deposition, the annual accumulated wet deposition of oxidised nitrogen (WNOx = $HNO_3$ + particulate nitrate + HONO + organic nitrates (e.g. PAN), for some models), reduced nitrogen (WNHx = $NH_3$ + particulate ammonium) and sulfur (WSOx = $SO_2$ + $H_2SO_4$ + particulate sulfate) was calculated from the modelled monthly estimates for the grid cells containing the measurement sites. In addition, model estimates of seasonal and annual accumulated precipitation, annual mean atmospheric

concentrations of total nitrate (TNO3 = $HNO_3$ + $PM_{10}$ nitrate), total ammonium (TNH4 = $NH_3$ + $PM_{10}$ ammonium) and total sulfate (TSO4 = $SO_2$ + $PM_{10}$ non-sea-salt sulfate) and annual accumulated dry deposition of oxidised (DNOx) and reduced (DNHx) nitrogen and sulfur (DSOx) were used to aid the interpretation of wet deposition estimates. For sulfate, all models except CMAQ corrected the model estimates to remove the contribution from sea-salt sulfate.

## 2.2 Observations

Estimates of accumulated seasonal and annual WNOx, WNHx, WSOx (non-sea-salt) and precipitation (at the same sites as the wet deposition) from the EMEP network over the period 1990–2010 were used to evaluate the model estimates. The seasonal and annual wet deposition was estimated by multiplying the volume-weighted mean concentration in precipitation by the total precipitation in the period. The concentrations for days with missing precipitation data were assumed to be equal to the volume-weighted average of the period (Hjellbrekke, 2017). For the evaluation of modelled atmospheric concentration

estimates, the EMEP network data of mean annual concentrations of total nitrate, ammonium and non-sea-salt sulfate were used. Although data are available for the individual gas and particulate species for many sites, the filter pack measurement methods used do not reliably estimate the partitioning of the gas and particulate N species and, therefore, the total (gas plus particulate) is used for the evaluation. Sites were selected that had data for at least 75% of the year and had valid data for at least 75% of the period 1990–2010, resulting in 39 sites for WNOx, 38 sites for WNHx, 36 sites for WSOx, 13 sites for

TNO3, 16 sites for TNH4 and 20 sites for TSO4 (Fig. 1 and Table S2). In order to compare the trends for the two eleven-year periods, a consistent set of sites was used that have data from the full twenty-one-year period. However, this approach led to gaps in the spatial coverage of observations (particularly in SW Europe) and so an additional analysis was carried out using all available sites that met the selection criteria for the period 2000–2010. Note that the availability of observations for several components is highly biased to certain parts of Europe. For example, total TNO3 and TNH4 concentrations are

mainly available for northern Europe and have very little overlap with wet deposition sites in the centre and west of the domain. It must also be noted that the evaluation of precipitation estimates was only done at the sites with observations of





wet deposition in order to assess the influence of model performance for precipitation on model performance of wet deposition. The aim was not to carry out a thorough evaluation of precipitation estimates, which would require a more detailed evaluation dataset, such as E-OBS (Haylock et al., 2008).

## 2.3 Model evaluation

The modelled wet deposition, precipitation and atmospheric concentration estimates were statistically evaluated using the package "openair" (Carslaw and Ropkins, 2012) for R (v3.3.2; R Core Team, 2016). Six metrics (as proposed by Chang and Hanna (2004)) were used to assess model performance: fraction of model estimates within a factor of two of the observed values (FAC2), fractional bias (FB), geometric mean bias (MG), normalised mean square error (NMSE), geometric variance (VG) and the Pearson correlation coefficient (r) (Table 1). The modStats function in openair was modified to include the metrics FB, MG NMSE and VG. Note that the convention of positive values of FB and values of MG > 1 was used to indicate model overestimation. Model evaluation was carried out for the full time series 1990–2010 for the five models that simulated all years and for the years 1990, 2000 and 2010 for all models. Model performance metrics were also compared with the acceptability criteria of Chang and Hanna (2004): FAC2 ≥ 0.5, |FB| ≤ 0.3, 0.7 ≤ MG ≤ 1.3, NMSE ≤ 1.5 and VG ≤ 4.

The observed and modelled trends in deposition and their significance were estimated using three methodologies: Mann-Kendall (MK), seasonal Mann-Kendall (SMK) and partial seasonal Mann-Kendall (PSMK) (see Appendix A). Since the observed and modelled trends can be non-significant and modelled and observed trends at a site can be in opposite directions, a statistical evaluation of the trends is not as straightforward as that for annual wet deposition. The modelled trends were evaluated using a two-step process. First, we assessed how well the models can classify the trends as significantly increasing, significantly decreasing and non-significant and then for the significant trends correctly classified by each model we assessed whether the magnitude of the trends were statistically similar to the observed trends (have overlapping 95% confidence intervals) or whether the model over- or underestimated the trends (non-overlapping 95% confidence intervals).

## 2.4 Attribution analysis

The EURODELTA Trends modelling experiment specifically included simulations that can be used to determine the contribution of several factors (changes in emissions, boundary conditions and meteorology) to the overall trends, as described in detail by Colette et al. (2017b). The methodology assumes that the overall trend ($\tau_{overall}$) is a linear composition of the trends due to changes in emissions ($\tau_{emissions}$), boundary conditions ($\tau_{boundary\ cond.}$) and meteorology ($\tau_{meteorology}$) plus a residual interaction term:

$$\tau_{overall} = \tau_{emissions} + \tau_{boundary\ cond.} + \tau_{meteorology} + Residual \qquad (1)$$



To calculate the contributions of each term to the overall trend for an 11 year period would require $11^3$ annual simulations, which is too demanding in terms of computing resources. Given their limited interannual variability, $\tau_{emissions}$ and $\tau_{boundary\ cond.}$ can be approximated as the difference in wet deposition over the eleven year period for simulations where the other factors are kept constant, divided by ten (to obtain the mean annual trend). For example, the change in wet deposition over the period 1990–2000 due to changes in emissions can be calculated from two simulations with emissions for 1990 and 2000, both with meteorology and boundary conditions for 2000. The choice of year for the factors that are held constant is arbitrary although the variability due to the year chosen has been shown to be less than a factor of ten smaller than the calculated change (Colette et al., 2017b). The overall trend is simply the trend calculated from the full model time series and the trend due to changes in meteorology is the trend calculated from a series of simulations with constant emissions (for the year 2010) minus $\tau_{boundary\ cond.}$. The residual term is calculated from the other terms in Eq. 1. This attribution analysis was done for the five models that carried out the required simulations (CHIM, EMEP, LOTO, MATCH and MINNI) and was applied to the sites with observations and to all model grid cells averaged over the nine sub-regions (Fig. S2 in the supplementary material) used by Colette et al. (2017b), which are based on the regional climatic zones originally defined in the PRUDENCE project (Christensen and Christensen, 2007).

## 3 Results

### 3.1 Emission Trends

Land-based $NO_x$ emissions for the period 1990–2000 decreased over most of the domain (Fig. S3) with the exception of the Republic of Ireland and southern parts of the domain (e.g. Portugal, Spain and Turkey). The largest decreases in reported emissions (more than 2000 kg km$^{-2}$ yr$^{-1}$) occurred in parts of Russia, Ukraine, Germany and the UK. Shipping emissions of $NO_x$ increased by more than 1 kg km$^{-2}$ yr$^{-1}$ over most of the domain. By contrast, for the period 2000–2010, most of the trends in $NO_x$ emissions in the east of the domain were not significant. For this period, the largest emission decreases (more than 2000 kg km$^{-2}$ yr$^{-1}$) occurred in the western part of the domain (Germany, Spain, France and the UK). Emission trends for shipping were either small or not significant for this period both over the Mediterranean Sea and the Atlantic Ocean.

European $NH_3$ emissions decreased during the period 1990–2000, mainly in response to the end of the communist system in Eastern Europe and the resulting structural changes (Sutton et al., 2003). Emissions decreased by more than 5% per year in seven countries (Bulgaria, Russia, Ukraine, Lithuania, Moldova, Estonia and Latvia). The largest decreases in reported emissions (> 100 kg km$^{-2}$ yr$^{-1}$) occurred in the Netherlands, NW Germany and Ukraine. Emission trends in the rest of the domain were mostly small or insignificant, apart from some significant increases in the Republic of Ireland, Spain, Turkey and north Africa. For the period 2000–2010, changes in $NH_3$ emissions were mostly not significant or within the range -100 to -5 kg km$^{-2}$ yr$^{-1}$.



Land-based $SO_x$ emissions decreased by more than 5 kg km$^{-2}$ yr$^{-1}$ for most of the domain in the period 1990–2000. Notable exceptions to this are the mostly non-significant trends in Greece and Spain and increasing trends in Turkey and northern Africa. Shipping emissions increased by more than 5 kg km$^{-2}$ yr$^{-1}$ for some Atlantic and Mediterranean shipping routes. Terrestrial $SO_x$ emissions also decreased in most of the domain for the period 2000–2010, although the decreases were

generally smaller than those of the previous decade. Shipping emission trends were mostly not significant for this period. The relative changes in emissions (Fig. S4) have a similar spatial distribution to the absolute trends although they highlight the large relative increases in emissions in some parts of the domain (e.g. marine areas for $NO_x$ and $SO_x$ and northern Africa for all compounds). Total domain emissions for $NO_x$, $NH_3$ and $SO_x$ decreased, on average, by 2.5, 1.9 and 5.4% per year, respectively, for the period 1990–2000 and by 1.7, 0.6 and 3.7% per year, respectively for the period 2000–2010.

## 3.2 Spatial distribution of modelled precipitation and wet deposition in 1990, 2000 and 2010

In order to analyse the spatial distributions of modelled precipitation and wet deposition and provide a basis for the subsequent discussion of the trends for the two eleven-year periods (1990–2000 and 2000–2010), this section analyses the spatial distributions of precipitation and wet deposition "snapshots" for the years 1990, 2000 and 2010.

The four meteorological models estimated similar spatial distributions of precipitation for 1990, with the largest precipitation amounts on the western and north-western coasts of Norway, the western coast of Scotland, the southern coast of Iceland and the Pyrenees and Alps mountain ranges (Fig. S5). The meteorological model used by the MATCH simulations estimated the largest domain-mean precipitation while that used for the CMAQ simulations estimated the smallest. For the year 2000 the models estimated similar distributions to those for 1990 although there was a noticeable shift southwards with less

precipitation on the Norwegian coast and more in the Iberian Peninsula and central parts of the domain (the Alps, Italy, eastern Adriatic coast and the Carpathian Mountains). Domain-mean precipitation differed very little between the two years with the largest difference estimated by the LOTO meteorological driver (7% increase). The southwards shift in precipitation continued between 2000 and 2010. The domain-mean concentration also differed very little between 2000 and 2010, with most meteorological drivers estimating a difference of less than 5%. The exception was the CMAQ meteorological driver

which estimated 23% more precipitation in 2010 compared with 2000. For 1990 MINNI estimated the smallest domain-mean wet deposition of oxidised nitrogen (WNOx) and MATCH the largest. (Fig. S6). However, in the east of the domain, EMEP estimated higher deposition than the other models. Despite the differences between the models, all of them estimated the highest WNOx in the centre and east of the domain, especially on the northern and southern slopes of the Alps, the southern coast of Norway and western Ukraine (corresponding mostly to

areas with large precipitation amounts). These deposition "hotspots" vary from model to model, with LOTO and MATCH, for example, estimating higher deposition rates on the southern slopes of the Alps compared with the northern slopes,





whereas EMEP and CMAQ estimated similar rates on both sides of the mountain range. These differences appear to be due to the spatial distribution of precipitation estimated by the meteorological driver. The spatial distributions of modelled WNOx estimates for 2000 are very similar to those for 1990 with a general decrease in deposition as a result of $NO_x$ emission reductions, especially in the east of the domain, reflecting the larger relative emission reductions in that region. The difference in domain-mean wet deposition between 1990 and 2000 was a decrease of between 13 and 20%. The models estimated a similar spatial distribution of WNOx for 2010 as for 2000 although domain-mean deposition decreased by 14 to 24%.

Similarly to WNOx, most of the models estimated the largest values of WNHx in 1990 for the slopes of the Alps, as well as for the Netherlands and NW Germany (Fig. S7), a well-known $NH_3$ emission hotspot (Sutton et al., 2013). The exception is LOTO, which did not estimate large values for the latter area. CHIM estimated the lowest mean WNHx and MATCH the highest. Also, similarly to WNOx, all models estimated a reduction in WNHx between between 1990 and 2000 for the east of the domain. However, the change in the domain-mean deposition varied between models with CHIM and LOTO estimating increases of 10% and 2%, respectively and the other models estimating decreases of 9 to 19%. Between the years 2000 and 2010, CHIM and LOTO estimated changes in domain-mean concentrations of 2% and -3% respectively whereas the other models estimated decreases of 10 to 17%.

The largest differences between the models, both in terms of the range of values and the spatial distributions was found for WSOx, with EMEP estimating the largest mean values in 1990 and CHIM the lowest (Fig. S8). CHIM, EMEP, LOTO and MINNI estimated the highest WSOx in NW Germany, whereas CMAQ estimated the largest values for the western coast of Norway (probably due to the inclusion of sea-salt sulfate). MATCH, on the other hand, estimated the highest deposition in Bulgaria in the southeast of the domain. In addition, both EMEP and MATCH estimated large values close to the active volcano Etna on the island of Sicily (Italy), as a result of the volcanic emissions included in these models. The spatial distributions of WSOx estimates for 2000 are similar to those of 1990, albeit with considerably lower values as a result of the large emission reductions within the domain. Domain-mean WSOx decreased between 32% and 48% for all models. The models estimated smaller decreases in the domain-mean WSOx between 2000 and 2010 (25–38%), with decreases mostly in the north and west of the domain.

### 3.3 Evaluation of model wet deposition estimates

Over the 1990–2010 period, all six models estimated a decrease in the wet deposition of oxidized nitrogen (WNOx) and sulfur (WSOx), when averaged (median) over all measurement sites (Figs. 2a and 2c and Figs. S9–S14). The model results for WNOx and WSOx follow the same pattern as the observed values, which also decreased, on average, over the same period. By contrast, the models estimated fairly constant rates of wet deposition of reduced nitrogen (WNHx) (Fig. 2b) over the same period whilst the median observed deposition decreased substantially between 1995 and 1996 and then remained



fairly constant. This decrease occurred at several sites although the largest influence came from two sites in France (FR0008R and FR0010R in Fig. S12). With regards to the variability between models, the estimates of WNOx are, on average, of a similar magnitude to the observed values, with the exception of MINNI, which underestimated deposition by more than a factor of two. For WNHx, EMEP and MATCH estimated similar values to those observed whereas CHIM,

CMAQ, LOTO and MINNI tended to underestimate them throughout the time series. CMAQ and LOTO estimated similar values of WSOx to those observed, whereas EMEP and MATCH tended to overestimate deposition and CHIM and MINNI tended to underestimate it.

Figure S15 shows the scatter plots of modelled vs. observed WNOx, WNHx and WSOx for the years 1990, 2000 and 2010 and Table S3 shows the performance evaluation of the six models for each of the three deposition components (WNOx,

WNHx and WSOx). Model performance is shown in Fig. 3 by plotting VG against MG for each model using a different symbol to indicate whether the acceptability criterion for FAC2 is met. The minimum value of VG for a given value of MG ($VG_{min} = \exp((\ln MG)^2)$) is also plotted. The metrics MG and VG were used for this since they are more suitable than linear measures such as FB and NMSE for distributions spanning many orders of magnitude (Chang and Hanna, 2004). For WNOx, five of the models (CHIM, CMAQ, EMEP, LOTO and MATCH) met all three of the performance criteria with

geometric mean biases (MG) ranging from 0.71 to 1.20 (corresponding to an underestimate of the geometric mean deposition of 29% and an overestimate of 20%, respectively). MINNI, on the other hand, underestimated the geometric mean by more than a factor of three (a MG of 0.30). The estimates of the MATCH model had the highest spatio-temporal correlation and those of MINNI the worst (r=0.84 and 0.65, respectively). EMEP and MATCH performed best for WNHx, meeting all three criteria. CMAQ and LOTO underestimated the geometric mean WNHx by 37% and 34%, respectively but

still met two of the three performance criteria whereas CHIM and MINNI underestimated the geometric mean deposition, by 55% and 60%, respectively. The WNHx estimates of the EMEP model had the highest spatio-temporal correlation and those of CHIM the worst (r=0.78 and 0.45, respectively). For WSOx, CMAQ and LOTO met all three criteria. EMEP and MATCH overestimated the geometric mean by 34% and 41%, respectively but still met two of the criteria. MINNI also met two of the criteria despite underestimating the geometric mean WSOx by about a factor of two. CHIM failed to meet any

criteria due to an underestimation by more than a factor of three. The WSOx estimates of the MATCH model had the highest spatio-temporal correlation and those of CHIM the worst (r=0.83 and 0.55, respectively).

### 3.4 Modelled and observed wet deposition trends

The partial seasonal Mann-Kendall (PSMK) trend calculations gave more significant trends than the other two methods (MK and SMK) for most models, periods and deposition components (Fig. S16). On average, this method gave significant trends

for 57% and 67% of the observed and modelled time series, respectively, compared with 40% and 52% for MK and 45% and 56% for SMK. Figs. S17 and S18 show that the absolute and relative trends calculated using the MK and SMK methods are



similar, although there is some scatter. The only difference between the SMK and PSMK methods is the calculation of significance and so the trends calculated by these two methods are the same. Since the PSMK method gave the most significant trends, the following analyses use the trends calculated using this method.

Fig. 4 shows the proportion of increasing and decreasing modelled and observed trends for the three wet-deposited compounds over the two eleven-year time periods and the full twenty-one-year period and whether the trends are significant ($p<0.05$). For WNOx, more significant decreasing observed and modelled trends were found for the second time period compared with the first. By contrast, the majority of observed and modelled trends of WNHx are not significant for both time periods although there are more increasing trends (both significant and non-significant) estimated by the models in the first period. Most of the observations and modelled estimates of WSOx have decreasing trends with a similar level of significance for both time periods and a higher proportion of significant trends than for both WNOx and WNHx. All three deposition components have more significant trends for the twenty-one-year period than for the two eleven-year periods and all sites have significant decreasing observed and modelled trends for WSOx for the longer time period.

With regards to the spatial distributions of the trends, most of the statistically significant observed trends of WNOx (both increasing and decreasing) for the period 1990–2000 are located in the central and north-eastern parts of the domain (Fig. 5). The five models estimated the most significant trends (mostly decreasing) in the east of the domain, although most of this part of the domain is not covered by the observations. These trends reflect the large reported emission reductions in Ukraine, Russia and Moldova but may have been moderated by increasing trends in precipitation in this region (Fig. S19). The models, however, failed to capture the significant observed increasing and decreasing trends in the centre of the domain (Fig. 5). Although there were also large decreases in reported emissions in the centre and west of the domain (e.g. Germany and the UK), the models did not estimate significant deposition trends in these regions, probably as a result of offsetting by increasing shipping emissions. CHIM estimated the largest area of significant trends (48% of domain), whereas MINNI estimated the smallest (24%). For the period 2000–2010, the majority of the statistically significant observed trends (mostly decreasing) are located in the central and western parts of the domain. The models also reproduce this western shift in significant trends, reflecting the spatial shift in decreasing emission trends and the lack of significant trends in shipping emissions (Fig. S3). Increasing observed and modelled trends in precipitation were also found for this region, which may have enhanced the deposition trends. Similarly to the first eleven year period, CHIM estimated the largest area of significant WNOx trends (48% of domain), whereas MINNI estimated the smallest (30%).

For WNHx during the period 1990–2000, the observations show significant trends (all but one decreasing) across the domain, with the largest decrease in the centre whereas the models did not estimate significant decreasing trends in this region (Fig. 6). All five models estimated the most significant decreasing trends in the east of the domain, corresponding to the region with the largest emission reductions but with poor coverage by observations. MATCH estimated the largest WNHx reductions for this period. All models estimated significant increasing trends around the English Channel despite





there being no emission increases in this area. This increase in WNHx is probably the result of increasing trends in precipitation in the region (Fig. S19) but could also be due to increased $SO_x$ and $NO_x$ emissions from shipping, which would enhance the production of particulate ammonium. Since the particulate form ($NH_4^+$) is less efficiently dry-deposited than the gaseous form ($NH_3$) (Duyzer, 1994), this could lead to a higher proportion of reduced N being wet-deposited. MATCH

estimated the largest area of significant trends (40% of domain), whereas LOTO estimated the smallest (21%). For the period 2000–2010, only four observed trends are statistically significant (three decreasing and one increasing) compared with 15 for the previous period. This decrease in trend significance is also present in the model estimates, which have fewer significant trends for land grid cells than the first eleven-year period. This reflects the smaller total domain emission decrease for the second period (1.0% $yr^{-1}$) compared with the first (1.6% $yr^{-1}$).

Most of the observed WSOx trends for the period 1990–2000 are significant decreasing trends (Fig. 7). The models also estimate significant decreasing trends in the regions represented by the observations and estimate the largest decreasing trends in the central and eastern parts of the domain (corresponding to the regions with the largest reductions in emissions). EMEP estimated the largest trends and the largest area of significant trends (72%) and CHIM the smallest trends and smallest area (54%). Similarly, for the period 2000–2010, all but one of the significant observed trends are decreasing and

are distributed throughout the area covered by the observations, with the exception of the northeast of the domain. The models, in general, estimated significant decreasing trends in the central and western parts of the domain. All models estimated small or non-significant trends in the south and southeast of the domain corresponding to the regions with increasing trends in modelled precipitation (Fig. S19). This suggests that the increasing precipitation partially offset the reduction in deposition in these regions during this period. LOTO estimated the largest area of significant trends (70% of

domain), whereas CHIM estimated the smallest (50%).

Focusing on the sites with observations, the observed trends of WNOx (mostly decreasing) were larger, on average, for the 2000–2010 period than for 1990–2000 (Fig. 8, top row). All of the models except CHIM were able to reproduce this difference. For WNHx, there were more decreasing observed trends during the first eleven year period than during the second. By contrast, all five models estimated more decreasing trends during the second period. However, there were very

few significant observed or modelled WNHx trends. This is not the case for WSOx, for which most of the observed and modelled trends were significant. Observed trends of WSOx (mostly decreasing) are largest, on average, during the first eleven year period. Although the models reproduce this difference, there is substantial variation between the models, with EMEP and MATCH estimating larger trends, on average, than those observed for the first period, CHIM and MINNI estimating smaller ones and LOTO estimating similar trends. This reflects the tendencies of the models to under- or

overestimate annual wet deposition. The trends calculated for the period 2000–2010 using all the available sites for that period are also shown in Fig. 8. Using all sites gives slightly smaller average observed and modelled trends for WNOx, WNHx and WSOx than using the same sites as the period 1990–2000 (i.e. less sites). This is probably due to the inclusion of





sites in the southeast of the domain for which the meteorological models estimated increasing precipitation trends for this period. Despite these small differences, the distribution of trends is very similar and we can conclude that the sites used in the trend analysis for both eleven year periods are fairly representative of the area covered by all sites.

Plotting the distributions of relative trends makes it possible to compare emission trends with observed and modelled

deposition trends (Fig. 8, middle row). Total $NO_x$ emissions in the domain decreased by 2.5% $yr^{-1}$ for the first period and by 1.7% $yr^{-1}$ for the second, whereas the average (median) observed trend for WNOx was -0.3% $yr^{-1}$ for the first period and -1.9% $yr^{-1}$ for the second. Modelled WNOx trends followed the same pattern as the observations, with average trends in the range -0.9 to -1.4% $yr^{-1}$ for the first period and -1.8 to -2.9% $yr^{-1}$ for the second, with the exception of CHIM, which gave larger trends, on average than the observations and other models for the first period (-2.1% $yr^{-1}$).

Total $NH_3$ emissions in the domain decreased by 1.6% $yr^{-1}$ for the first period and by 1.0% $yr^{-1}$ for the second. The average observed trend for WNHx for the two periods also followed this pattern with a larger decrease for the first period. However, the average modelled trends for the first period were close to zero for three out of the five models. The exceptions being CHIM, which estimated an average trend of +1.5% $yr^{-1}$ and MATCH, which estimated an average trend of -1.3% $yr^{-1}$. Both observed and modelled average trends for the second period were in the range 0.3–1.8% $yr^{-1}$ (decreasing). Total $SO_x$

emissions in the domain decreased by 5.7% $yr^{-1}$ for the first period and by 4.5% $yr^{-1}$ for the second. The observed and modelled trends for WSOx also followed this pattern with larger average trends during the first period (3.7–5.1% $yr^{-1}$) compared with the second (3.6–4.7% $yr^{-1}$), with the exception of LOTO, which estimated similar average trends for both periods (ca. 5.0% $yr^{-1}$). The use of relative trends instead of absolute trends reduce the differences between the models and between the models and the observations for all three components and both time periods as a result of removing systematic

biases in the models. For the simulations with emissions held at the 2010 values (Fig. 8, bottom row), the models predominately estimated increasing trends of WNOx, WNHx and WSOx for the first period and decreasing trends for the second period suggesting that the changes in meteorology and/or boundary conditions also influenced the trends in wet deposition. In fact, the modelled median deposition trends can be approximated by summing the relative emission trends and the relative deposition trends from the constant emission scenarios (Fig. S20), with the exception of the positive WNHx

trends for the period 1990–2000, probably due to the $SO_x$ and $NO_x$ emissions from shipping, as discussed above. The contribution of the changes in meteorology and/or boundary conditions to the modelled trends is investigated further in the attribution analysis.

### 3.5 Evaluation of modelled wet deposition trends

As shown in Fig. 4, most of the observed trends in WNOx for the two eleven year periods were not significant. For the first

eleven year period, all but one of the models classified the trends correctly for the majority of the sites (Fig. 9). The exception is CHIM, which only correctly classified 41% of the observed trends because it estimated more significant trends



than the other models (see Fig. 4). Model performance was similar for all models for the second period, with all models correctly classifying between 35-53% of the observed trends. The models failed to classify correctly more than 60% of the non-significant trends since they estimated more significant trends than those observed. For the full twenty-one-year period all models correctly classified the significant trends but only MINNI classified correctly some of the non-significant ones.

The models classified correctly 39-52% of the WNHx trends (mainly the non-significant ones) during the first period. Model performance was better for the second eleven year period but this improvement was due to a larger number of non-significant observed trends. Model performance was more variable for the twenty-one-year trends in WNHx, with CHIM correctly classifying the least trends (26%) and MATCH the most (65%). Overall, model performance was best for the trends in WSOx, with the models correctly classifying 82-91% and 70-79% of the trends during the first and second eleven-year

periods, respectively and all of the trends for the twenty-one-year period. Although there were some significant increasing trends in observed wet deposition of WNOx, WNHx and WSOx for some of the periods, none of the models correctly classified them.

Of the few significant observed decreasing trends in WNOx correctly classified by the models (blue bars in Fig. 9) most of them were either underestimated by the models or were statistically similar to the observed trends (i.e. have overlapping

confidence intervals) for both eleven-year periods (Fig. S21). MINNI underestimated all but one of the trends, reflecting the fact that this model underestimated annual wet deposition estimates. Similarly for WNHx, the models tended to estimate smaller or similar trends to those observed although the number of correctly classified trends was small ($\leq 2$ for each period). For WSOx there are more significant observed trends and, as a consequence, more significant trends correctly classified by the models. Model performance was similar for both eleven-year periods with some models underestimating or estimating

similar trends to those observed at most sites (CHIM, LOTO and MINNI) and other models overestimating or estimating similar trends for most sites (EMEP and MATCH). Since most of the observed and modelled WSOx trends were significant, it is possible to statistically evaluate the modelled trends using the metrics FAC2, FB, NMSE and r (MG and VG are not valid for negative values). Figure 10 (and Table S4) shows that CHIM and MINNI underestimate the absolute trends, on average, by a factor of about four and two, respectively. Model bias for trends reflects the model bias for wet deposition with

CHIM, LOTO and MINNI underestimating and EMEP and MATCH overestimating. Model performance is better for the relative trends, with all models meeting the acceptability criteria. The twenty-one-year trends were not evaluated since the majority of the trends had a non-linear character (i.e. the trends of the two eleven-year periods were significantly different; $p < 0.05$) and the linear trend estimation methods were not valid.

### 3.6 Trend attribution analysis

Figure 11 shows the contributions of the changes in emissions, boundary conditions and meteorology to the modelled trends of WNOx, WNHx and WSOx at all measurement sites. For all three deposition components and both time periods, the



largest contribution to the overall modelled trend is the reduction in emissions. Many of the overall trends are smaller than the trends due to emissions alone as a result of positive contributions of changing meteorology and non-linear interactions (which also could include contributions from the meteorology). However, for most of the trends, this offsetting is smaller for the second period resulting in a stronger influence of the emission reductions for this period. The larger offsetting by meteorology and other interactions (represented by the residual component) for the first period can also be seen in the regional analysis of the land grid cells presented in Figs. S22–24, especially for England (EN) and Mid-Europe (ME). This difference in offsetting between the periods is not as apparent for the analysis of the land grid cells of the entire domain since the offsetting is larger in the second period for some regions, such as the Iberian Peninsula and the Mediterranean, which are poorly represented by the observations (only one site).

### 3.7 Evaluation of precipitation estimates

Since precipitation rates have a strong influence on wet deposition, it is useful to evaluate model performance for precipitation at the same sites with observations of wet deposition to see if it can help to explain model performance for WNOx, WNHx and WSOx. Model biases are very small for accumulated annual precipitation, with three meteorological models (those used by CHIM, CMAQ, EMEP, LOTO and MINNI) underestimating the geometric mean precipitation (by 4-8%) and one overestimating it (that used by MATCH, by 5 %) (Fig. 12 and Table S5). Model biases are also small for seasonal precipitation. The meteorological models used by all of the CTMs except MATCH performed worst in summer with underestimations of 18-28%. By contrast, the meteorological model used by MATCH had a very small bias (2%) for this season. A comparison of the observed precipitation trends for the two eleven-year periods shows that the trends are small and positive, on average, and very similar for the two periods, although the average trends for the first period are slightly larger than those for the second (Fig. 13). CHIM, EMEP and MINNI estimated very similar median trends to those of the observations. The HIRLAM model used by MATCH, also gave trends in the same range although this model estimated slightly larger median trends for the second period compared with the first. By contrast, RACMO2 (used by LOTO) estimated larger median trends than the other models and estimated positive median trends for the first period and negative for the second, which could be due to the fact that the RACMO2 simulation is not nudged towards the observed precipitation. Very few (<10%) of the observed and modelled precipitation trends were statistically significant.

### 3.8 Evaluation of atmospheric concentration estimates

Since wet deposition estimates are also strongly dependent on atmospheric concentrations in the column, it is useful to evaluate model performance for (surface) concentrations to see if it can help to explain model performance for wet deposition. A more detailed analysis of the trends in atmospheric concentrations estimated by the EDT simulations is provided by Ciarelli et al. (2018). In contrast to wet deposition, for which most models underestimated deposition rates or





had a small bias (with the exception of EMEP and MATCH for WSOx), all models overestimated mean atmospheric concentrations of TNO3, TNH4 and TSO4 or had a small bias (Figs. 14 and S25 and Table S6). All models overestimated the geometric mean TNO3, with CMAQ overestimating the most (by 115%) and MINNI the least (by 25%). CMAQ also overestimated the geometric mean TNH4 the most (by 47%) whereas the estimates by EMEP and MATCH had a small

negative bias (4-17%). All models overestimated geometric mean TSO4 concentrations, with CMAQ overestimating the most (by 114%) and the EMEP and LOTO the least (by 19-20%). The overestimation of TSO4 by CMAQ is probably due to the inclusion of sea-salt sulfate in this model. One way to investigate whether the underestimation of wet deposition (at the sites where concentrations were measured) and the overestimation of concentrations are linked is to see if the performance statistics of the two quantities are correlated (Table S6). A correlation analysis shows that, in general, the statistics are not

correlated except for the MG of WNHx and TNH4, which are negatively correlated (r=-0.78), i.e. a tendency for the more the model underestimates wet deposition the more it overestimates atmospheric concentrations. This suggests that, at least for reduced nitrogen, an underestimation of wet deposition could be leading to an overestimation of TNH4 concentrations. However, the models tended to overestimate wet deposition more than they underestimated concentrations so there could be other factors involved, such as removal by dry deposition.

3.9 **Analysis of dry and total deposition**

Although there are no observations for evaluating dry deposition it is still useful to compare the dry deposition estimates of the models at the same sites that were used for the evaluation of wet deposition in order to determine whether the differences between the estimates of dry deposition can explain the differences between the estimates of wet deposition. Fig. 15 shows that for dry deposition of oxidised N (DNOx), the median model estimates differ by a factor of about two for most of the

time series with LOTO estimating the lowest rates and CMAQ the highest. These high DNOx estimates by CMAQ could be due to the high TNO3 concentrations estimated by this model. There is slightly more agreement between the models for dry deposition of reduced N (DNHx, with median estimates differing by about a factor of 1.5. However, MINNI estimated an increase in dry deposition between 1996 and 1999, which did not occur for the other models. Out of the other models, MATCH estimated the smallest values and CHIM the largest for most of the time series. The low DNHx estimates of

MATCH could be due to the low estimates of TNH4, which in turn could be the result of the small overestimation of WNHx by this model. For dry deposition of sulfur (DSOx), the estimated median deposition values differ by about a factor of two, mainly as a result of CHIM estimating higher values than the other models for the entire series. This could partly be due to the underestimation of WSOx by this model.

With regards to the total deposition (wet plus dry), MINNI and LOTO estimated smaller median values for oxidised nitrogen

than the other models by a factor of 1.5 to 2 for the entire time series, whereas CMAQ estimated the largest values for the years 1990 and 2000 (Fig. S26). CHIM and MINNI estimated the lowest values for reduced nitrogen and EMEP and



MATCH the highest for most of the time series, with a similar range of variability as for oxidised nitrogen. Summing the reduced and oxidised components to obtain total nitrogen deposition (Fig. S27), shows that MINNI and LOTO estimated the lowest median values (as a result of their low estimates of oxidised N deposition) and CMAQ and MATCH the highest (as a result of the CMAQs high estimates of oxidised N deposition and MATCH's high estimates for both the oxidised and

reduced components). For sulfur, EMEP and MATCH estimated very similar rates of total deposition, as did CMAQ for 2000 and 2010. CHIM and LOTO estimated similar rates of S deposition, which were lower than those of EMEP and MATCH whereas MINNI estimated the lowest values (Fig. S27).

## 4. Discussion

### 4.1 Uncertainties and limitations of the methods used

Like any study involving observed data and/or model simulations, the results presented here are subject to various sources of uncertainty. The national emission data used in the simulations are based on the officially reported values. The European Environment Agency suggests that the emission estimates for European member states have an uncertainty of about ±10% for $SO_2$, ±20% for $NO_x$ and ±30% for $NH_3$ (EEA, 2008). These values are consistent with those of Schöpp et al. (2005), who estimated that the national total emissions for 1990 used in the RAINS integrated assessment model for have an uncertainty

of ±6–23% for $SO_2$, ±8–26% for $NO_x$ and ±9–23% for $NH_3$, although the EEA estimate for the latter is larger. The spatial distributions of emissions estimated using source proxies are also subject to considerable uncertainty, which is difficult to quantify (Kuenen et al., 2014). The detail of the data used to generate these proxies also changes with time as more (and possibly better) information is made available due to improvements in methodologies or due to more stringent reporting commitments.

Another source of uncertainty is the meteorological data used in the simulations, as well as the procedures within the models that parameterise the atmospheric conditions from those data. Since annual wet deposition is correlated with accumulated precipitation, it seems logical to focus the discussion on precipitation. As shown in Fig. 12, the annual accumulated precipitation calculated by the meteorological models used in most of the simulations (based on ERA-Interim reanalysis data) is lower than the observed precipitation by 4-8%, on average. This is consistent with the analyses of Dee et al. (2011)

who showed that the ERA-Interim reanalysis precipitation data underestimates mean precipitation rates by 0.2–1.0 mm day$^{-1}$ for most of Europe with respect to the observation-based estimates of the Global Precipitation Climatology Project (Adler et al., 2003), although the latter is also subject to bias. Taking the average annual precipitation from the observations used in this study (945 mm), an underestimation of 0.2–1.0 mm day$^{-1}$ corresponds to an underestimation of 8–38%. In addition to the uncertainties in annual accumulated precipitation, the departure of the hourly, daily and monthly modelled precipitation from

the observed values could lead to large errors in the modelled wet deposition, although the net effect of these uncertainties is





not expected to be a large systematic under- or overestimation of wet deposition. Uncertainties in other meteorological variables such as wind speed, humidity and boundary layer heights may affect the vertical profiles of pollutants and cloud formation, both of which could lead to errors in the wet deposition estimates. There is also uncertainty in the boundary conditions used in the model, both in the hourly time series used and the long-term changes over time.

The spatial resolution used for the model simulations can also add uncertainty since the model estimate for a grid cell may not be representative of the location of the measurement site. The grid cell areas of the model domain range from approximately 425 to 1050 km$^2$, which may be larger than the representative areas of the individual sites. However, the use of the EMEP network, which aims to maximise the spatial representativeness of the measurement sites should help to minimise this uncertainty, although this may not be possible in mountainous areas with very spatially variable precipitation

patterns. Furthermore, sites close to farming areas may overestimate deposition of reduced nitrogen with respect to the average deposition within the grid cell. The observations used in this analysis are also subject to uncertainties resulting from the field measurement technique used, the laboratory analysis methods and the data processing (e.g. gap filling). The WNHx data from the two sites in France that contributed most to the decrease in average observed deposition between 1995 and 1996 were discussed with the site operators. An assessment of the sampling equipment and analytical techniques used did

not provide any reason to discard these data.

With regards to the trend analyses, the small number of measurement sites that show significant trends in the observed values (especially for WNHx) makes it difficult to perform a robust analysis to determine whether or not the models can reproduce the trends. Another limitation to the trend analysis is the requirement for linear trends, which is not the case for most of the trends for the period 1990–2010. A trend analysis for the twenty-one-year period using non-linear trend

estimation methods (see e.g. Venier et al., 2012) could provide a more robust evaluation. However the linear trend analysis does allow the assessment of trends for shorter periods provided there are enough sites with significant observed and/or modelled trends.

## 4.2 Model performance for wet deposition

Although the uncertainties highlighted above may contribute to the systematic underestimation of wet deposition by many of

the models, it is unlikely that they account for all of the model bias.  For example, in the worst case, an underestimate of 30% in the emissions and 8% in the precipitation is unlikely to give an underestimate of wet deposition by 70% (although it is not impossible) and so there are probably other explanations for the underestimations by some models. Also, for the same input data, some models have a very small bias while for others it is large. It also seems unlikely that the problem comes from underestimated emissions since this would be expected to also lead to an underestimate of atmospheric concentrations

(unless the errors are compensated by errors in other variables, such as the boundary layer height), which is not the case. Another possibility is an overestimation of dry deposition, which would leave insufficient pollutant in the atmosphere and so



wet deposition would be underestimated. However, this would also be expected to lead to an underestimation of atmospheric concentrations as well. The most plausible explanations for model bias are deficiencies in the wet deposition schemes of the models (e.g. uncertainties in the scavenging coefficients for gases and particles) and/or errors in the frequency and intensity of precipitation events, the vertical profiles of the pollutants or the parameterisation of clouds and

cloud chemistry. Similar conclusions were made by Vivanco et al. (2017), who found a general underestimation of wet deposition by several models for four campaigns over the period 2006–2009. A comparison of model biases between their study and ours for the models common to both studies (CHIM, CMAQ, EMEP, LOTOS and MINNI) shows that model performances in the two studies are fairly consistent. For example, in both studies MINNI underestimated WNOx the most, whereas EMEP had a very low bias and the models CMAQ, LOTO and MINNI underestimated WNHx the most and EMEP

had the smallest bias. Also, in both studies LOTO and MINNI tended to underestimate WSOx and EMEP tended to overestimate it. CHIM had differing results depending on the study (e.g. underestimating WNOx in this study but having a very small bias in Vivanco et al. (2017), but that may be because of the different model version and time periods used (annual simulations vs. month-long campaigns). Despite these differences, the results are sufficiently consistent for the conclusions made by Vivanco et al. (2017) to hold, namely that the tendency of models to underestimate wet deposition and

overestimate atmospheric concentrations (as is the case for oxidised and reduced nitrogen) is likely to be due to deficiencies in simulating wet deposition processes, which could be related to the vertical concentration profiles, scavenging coefficients or in-cloud processes, including the parameterization of clouds. The case of WSOx is slightly different. In this study, CHIM, CMAQ, LOTO and MINNI tended to underestimate WSOx and overestimate TSO4, which again could be due to deficiencies in the wet deposition processes (including vertical concentration profiles, cloud parameterisation, etc.) whilst the

other models (EMEP and MATCH) overestimated the wet deposition and the concentrations, which could be due to an overestimate of $SO_2$ concentrations at the rural locations of the measurements due to the coarse model spatial resolution, as suggested by Vivanco et al. (2017). In the present study, MATCH estimated higher wet deposition rates, on average, than EMEP for oxidised and reduced N (Table S3). This is consistent with the results of Simpson et al. (2014), who showed that MATCH estimated mean WNOx, WNHx that were 21% and 15%, respectively, higher than those of EMEP. Enghardt et al.

(2017) also found this for total deposition of oxidised and reduced N, concluding that the atmospheric lifetime of the considered species is longer in EMEP than in MATCH. Despite these clues as to why some models perform better than others, it is out of the scope of this study to investigate the reasons why individual models perform well or not-so-well.

The results presented here are also fairly consistent with studies that have evaluated individual models, despite the fact that these studies used different model versions, meteorological data and measurement sites. For example, Simpson et al. (2006)

found that the EMEP model underestimated mean WNOx and WNHx by 16–26% and 16–17%, respectively when compared with measurement data from 160 forest sites. In the present study, EMEP underestimated WNOx and WNHx by 2% and 14%, respectively. For WSOx, Simpson et al. (2006) found that the EMEP model also underestimated mean deposition by 9–



26%, whereas in the current study we found an overestimate by 31%. On a national level, Schaap et al. (2017) found that LOTOS-EUROS underestimated the mean wet deposition of oxidised and reduced nitrogen by 38% and 21%, respectively, and that of sulfur by 44%, when compared with observations made at 150 sites in Germany. In the present study LOTOS-EUROS also underestimated mean deposition by 35%, 41% and 23%, for WNOx, WNHx and WSOx, respectively.

**4.3 How well can the models reproduce the observed annual trends in N and S wet deposition for the period 1990–2010?**

Despite the limitations and uncertainties of the analyses presented here, it has been possible to statistically evaluate the modelled trends in deposition. With regards to the significance and direction of the trends, the models generally reproduce the observed larger and more significant decreasing WNOx trends in the second eleven-year period compared with the first, despite similar relative emission reductions for both periods. The analysis of precipitation trends, simulations with constant emissions and the trend attribution analysis all suggest that this is due to a larger increase in precipitation and/or other changes in the meteorology during the first period, partially offsetting the decrease in wet deposition due to emission reductions.  This effect can be seen more clearly in Fig. 16, which shows that the median relative trend of the observed and modelled WNOx at the measurement sites is smaller for the first period. In fact, all models estimate smaller average relative trends in wet deposition than those of the emissions during the first period and larger average relative trends during the second period due to changes in the meteorology and/or boundary conditions.  Another factor that could influence the different responses of wet deposition during the two periods to changes in emissions is the non-linear response of TNO3 concentrations to reductions in $NO_x$ emissions as a result of decreasing emissions of $SO_2$. Ciarelli et al. (2018) analysed the trends in atmospheric concentrations in the EDT dataset and found that the decreasing trends in $HNO_3$ concentrations were larger than those of particulate nitrate concentrations during the first eleven year period probably as a result of an increased availability of "free-ammonia" following the strong reduction in $SO_x$ emissions, thus causing a more efficient conversion of $HNO_3$ to the particle phase. This potential shift in the thermodynamic equilibrium of $HNO_3$ could influence the wet deposition trends due to differences in scavenging coefficients of the gas and particle species and the increased atmospheric lifetime of oxidised nitrogen. This increased lifetime could lead to increased export of TNO3 out of the domain resulting in a reduction of the trends in TNO3 and consequently WNOx, with respect to the emission reductions. However, as the simulations with constant emissions and the attribution analyses show, the influence of changing meteorology and boundary conditions can explain most of the differences between the two periods. Similarly for WSOx, the models reproduce the level of significance and direction of trends observed in the two periods. In this case, the models and observations suggest a larger average relative decrease in deposition for the first eleven year period (especially for MATCH), although the differences between the periods are not as large as the differences in relative emission trends for the two periods, again due to the partial offset during the first period as a result of changing meteorology and/or other factors.  The low level of trend significance for WNHx, on the other hand, results in very variable median trends, from which it is hard to draw conclusions.



Despite the ability of the models to generally reproduce the direction and significance of most of the observed trends (at least for WNOx and WSOx), the models tended to estimate significantly smaller or similar trends for WNOx, reflecting the tendency of most models to underestimate annual deposition. Similarly, the models that underestimated annual WSOx, on average, (CHIM, LOTO and MINNI) tended to estimate significantly smaller or similar trends to those observed whereas those that overestimated annual WSOx (EMEP and MATCH) tended to estimate significantly larger or similar trends to those observed. The use of relative trends reduces these systematic errors. The fact that the year-to-year relative changes in modelled deposition are more reliable than the absolute changes and that model biases do not change much over the 20 year time period (Fig. S28) opens up a possibility for improving model estimates of deposition. If the model bias (MG) is calculated for an initial period (e.g. the first three years of the time series), then the bias correction necessary to remove this initial bias (multiplying the model estimates by 1/MG) can be applied to the entire time series, thus reducing model error for the sites with observations. Bias-correcting the full time series in this way improves model performance for wet deposition considerably (Fig. 17). For this dataset, the bias-corrected data is fairly insensitive to the choice of the length of the initial bias calculation period (Fig. S29). Model trend estimates were also improved with this bias correction (Fig. S30), especially for WNOx and WSOx. This is useful since the method could be applied to future time series with the initial bias calculation period referring to the present period (e.g. the last three years with available observations).  However, due to the limited number of sites with available observations it is not possible to evaluate these bias-corrected estimates with observations that have not been used to calculate the bias correction. Despite this limitation, this approach has the potential to provide more robust predictions of future wet deposition rates by minimising the systematic errors of the models and also improve ensemble model estimates. Clearly, the focus should be on improving the models to reduce systematic biases but in the meantime this method provides a way of obtaining more reliable model estimates of wet deposition for future time series.

## 5. Conclusions

We have evaluated the wet deposition of sulfur (WSOx) and oxidised (WNOx) and reduced (WNHx) nitrogen estimated by six atmospheric chemistry transport models using observations from the EMEP monitoring network for the period 1990–2010. Most of the models met the pre-defined acceptability criteria for the three components, although there were some exceptions. MINNI underestimated the wet deposition of all three components by more than a factor of two to three, on average. The fact that all models used the same emissions, boundary conditions and, where possible, meteorology suggests that this general underestimation is due to the parameterisation of the model, such as deficiencies in the wet deposition scheme, the vertical concentration profiles of the pollutants or the parameterisation of clouds and cloud chemistry. The other exception is Chimere (CHIM), which underestimated WNHx and WSOx by more than a factor of two to three, on average. The fact this model had a small bias for WNOx suggests that the model underestimation of WNHx and WSOx is related to the parameterisations for reduced nitrogen and sulfur compounds, such as the species-specific scavenging coefficients, the



gas phase or cloud chemistry schemes or the aerosol physics. In order to understand the underestimation of wet deposition by MINNI and Chimere, a detailed study of the chemical and physical processes occurring in the model column would be required, which is out of the scope of the present study.

More than half of the observed trends of WNOx and WNHx for the two periods 1990–2000 and 2000–2010 were not significant, making it difficult to evaluate the modelled trends statistically. For the sites with both significant observed and modelled trends, the models tended to estimate similar or smaller trends than those observed, with MINNI underestimating all but two of the observed trends, reflecting the tendency for this model to underestimate WNOx and WNHx. Despite small but significant $NH_3$ emission reductions for most of the modelling domain during the first period, all of the models estimated increasing trends of WNHx near the English Channel. This is probably due to increased precipitation but could also be due to increased $NO_x$ emissions from shipping leading to an increase in particulate ammonium formation. More than 80% of the observed trends of WSOx were significant for both eleven year periods. MINNI and CHIM tended to estimate similar or smaller trends than those observed while EMEP and MATCH tended to estimate similar or larger trends and LOTO had a more balanced performance. The evaluation of the modelled absolute WSOx trends showed that EMEP, LOTO and MATCH met most of the acceptability criteria whereas CHIM and MINNI did not, underestimating the absolute trends, on average, by a factor of about four and two, respectively. This was a direct consequence of the consistent underestimation of WSOx by these models. The use of relative trends improved model performance greatly with all models meeting all acceptability criteria. This is a consequence of the fairly constant model biases for wet deposition, which makes it possible to improve the predictions of wet deposition for future scenarios by adjusting the model estimates using a bias correction calculated from past observations.

An analysis of the factors contributing to the modelled trends shows that reductions in emissions contribute most to the trend estimates. However, changes in meteorology, boundary conditions and other factors also have an influence in the trends estimated at monitoring sites, suggesting that the emission reduction measures had a larger effect during the second period at these sites. Changes in atmospheric chemistry due to large reductions in $SO_2$ emissions during the first period (Ciarelli et al., 2018) could also have influenced the wet deposition trends, although to a smaller degree. These factors will also influence the inter-annual variability of the observed wet deposition leading to a large number of non-significant trends. Even with reported emission reductions of the order of 10-25% during an eleven year period (as in the case of $NO_x$ and $NH_3$), significant trends (of WNOx and WNHx) are observed at less than half of the sites. Larger emission reductions are required in order to detect significant trends for an eleven year period at most sites, such as the 50-60% emission reductions reported for $SO_x$.



## Appendix A: Trend analysis

The statistical significance of the trends in observed and modelled wet deposition, as well as in the emissions was calculated using the Mann-Kendall (MK) test, which assesses whether there is a statistically significant monotonic trend in a data time series (Mann, 1945; Kendall, 1970). This is a non-parametric test and so is suited to datasets that are not necessarily normally distributed (unlike other methods, such as linear regression). The method can be used to assess the significance of a trend, even if it is non-linear, and is fairly insensitive to missing data. The statistic S is calculated as the difference between the number of pairs of increasing values in the time series (out of all pair combinations) and the number of pairs of decreasing values, with a positive sign if there are more increasing pairs and a negative sign if there are more decreasing pairs. For time series of more than ten values, S is assumed to be normally distributed and, therefore, its variance can be calculated. From the values of S and its variance, the test statistic Z can be calculated using standard statistical methods (Gilbert, 1987). A significant trend is defined as one that has a value of Z less than a pre-defined value (in our case 0.05 for a 95% confidence level). The magnitude of the trend was calculated following Sen (1968), as the median value of the slopes between all data pair combinations in the time series. Relative trends were calculated as the ratio of the Sen's slope (Q) to an estimate of the data at the start of the period, which was calculated as the median of the values $x_i - Qt_i$, where $x_i$ is the data for time step i and $t_i$ is the time elapsed since the start of the period (Salmi et al., 2002).

Since the temporal variability of wet deposition depends strongly on seasonal precipitation cycles, we also applied the trend analysis to the observed and modelled deposition for winter, spring, summer and autumn individually and then calculated the trend significance from the sum of the S values for each season and the Sen's slope as the median value of the slopes between all data pair combinations in the complete time series, using data pairs from the same season only (Hirsch and Slack, 1984). In order to further take into account the correlation of wet deposition with accumulated precipitation, we also used an extension of this seasonal Mann-Kendall (SMK) method, the partial seasonal Mann-Kendall (PSMK) method, which uses a co-variable (in this case precipitation) to estimate the trend significance (Libiseller and Grimvall, 2002). All trend analyses were carried out using the "rkt" function for R developed by Marchetto et al. (2013) and modified to calculate the relative Sen's slopes.

## Author contribution

ACo coordinated the EURODELTA-Trends (EDT) exercise and WA was responsible for the compilation and quality-control of the observations. The following modelling teams set-up, pre-processed, ran and post-processed the simulations for each model: FC, BB, MGV and ACo for CHIM; ST, MTP for CMAQ; HF and PW for EMEP; AM and MS for LOTO; CA and RB for MATCH; MM, MA, GB, ACa and MD for MINNI. Additional post-processing of model output and upload to the





AeroCom server was done by KC. All of the analysis presented in this paper was carried out by MT and MGV with assistance from GC, KM, NO, VR, YR and ACo.

**Data availability**

Technical details of EURODELTA project simulations that permit the replication of the experiment are available on the wiki
of the EMEP Task Force on Measurement and Modelling (https://wiki.met.no/emep/emep-experts/tfmmtrendeurodelta), which also includes ESGF links to corresponding input forcing data. The EURODELTA-Trends model results are made available for public use on the AeroCom server (https://wiki.met.no/aerocom/user-server).  See Colette et al. (2017a) for full terms and conditions for the use of these data.

**Competing interests**

The authors declare that they have no conflict of interest.

**Acknowledgements**

The GAINS emission trends were produced as part of the FP7 European Research Project ECLIPSE (Evaluating the Climate and Air Quality Impacts of Short-Lived Pollutants); grant no. 282688. The Ineris coordination of the Eurodelta-Trend exercise was supported by the French Ministry in charge of Ecology in the context of the Task Force on Measurement and
Modelling of the EMEP programme of the LRTAP Convention. Meteorological forcings with the WRF model were provided by R. Vautard and A. Stegehuis from LSCE/IPSL. We also thank Erik van Meijgaard of the Royal Netherlands Meteorological Institute (KNMI) for providing the RACMO2 simulations that were used by LOTOS-EUROS. We would like to express our thanks to all those who are involved in the EMEP monitoring efforts and have contributed through operating sites, performing chemical analysis and by submissions of data. This work was supported by the Co-operative
Programme for Monitoring and Evaluation of the Long-range Transmission of Air pollutants in Europe (EMEP) under the UNECE. The participation of CIEMAT was financed by the Spanish Ministry of Agriculture and Fishing, Food and Environment. The CHIMERE simulations where performed using the TGCC super computers under GENCI computing allocation. The MATCH participation was partly funded by the Swedish Environmental Protection Agency through the research program Swedish Clean Air and Climate (SCAC) and NordForsk through the research programme Nordic WelfAir
(grant no. 75007). The computing resources and the related technical support used for MINNI simulations have been provided by CRESCO/ENEAGRID High Performance Computing infrastructure and its staff. The infrastructure is funded by ENEA, the Italian National Agency for New Technologies, Energy and Sustainable Economic Development and by Italian



and European research programmes (http://www.cresco.enea.it/english). MINNI participation to this project was supported by the "Cooperation Agreement for support to international Conventions, Protocols and related negotiations on air pollution issues", funded by the Italian Ministry for the Environment, Land and Sea. Giancarlo Ciarelli was supported by ADEME and the Swiss National Science Foundation.

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



**Table 1: The six performance metrics relating model estimates (M$_i$) to the observed values (O$_i$) used to assess model performance.**

| Performance metric | Definition | Optimum value |
|---|---|---|
| Fraction of model estimates within a factor of two of the observations (FAC2) | $0.5 \leq \dfrac{M_i}{O_i} \leq 2.0$ | 1 |
| Fractional bias (FB) | $FB = \dfrac{2(\overline{M} - \overline{O})}{(\overline{M} + \overline{O})}$ | 0 |
| Geometric mean bias (MG) | $MG = \exp(\overline{\ln M} - \overline{\ln O})$ | 1 |
| Normalised mean square error (NMSE) | $NMSE = \dfrac{\overline{(O - M)^2}}{\overline{O}\,\overline{M}}$ | 0 |
| Geometric variance (VG) | $VG = exp\left[\overline{(\ln O - \ln M)^2}\right]$ | 0 |
| Pearson correlation coefficient (r) | $r = \dfrac{1}{(n-1)} \sum_{i=1}^{n} \left(\dfrac{M_i - \overline{M}}{\sigma_M}\right)\left(\dfrac{O_i - \overline{O}}{\sigma_O}\right)$ | 1 |

(a)          (b)          (c)

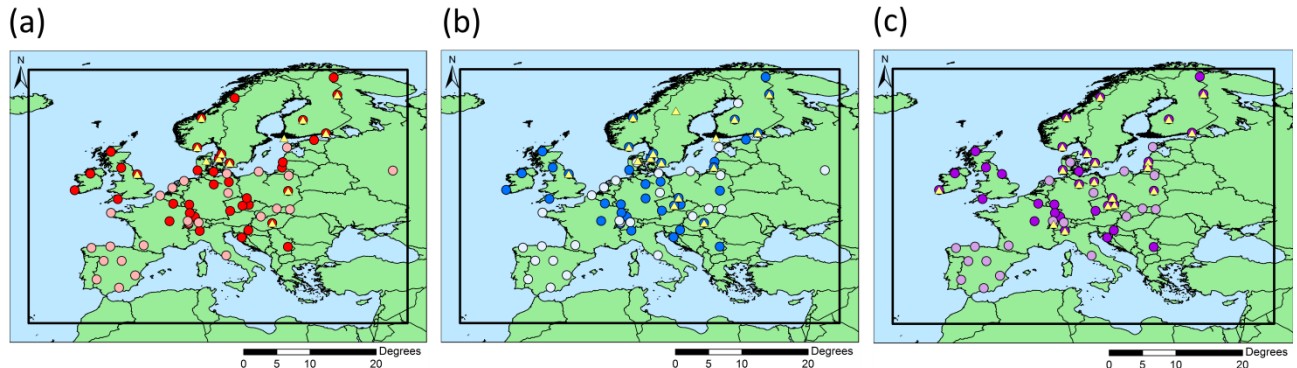

5 **Figure 1: Locations of the measurement sites used in the evaluation of wet deposition and atmospheric concentrations of a) oxidised N (WNOx and TNO3), b) reduced N (WNHx and TNH4) and c) sulfur (WSOx and TSO4). Dark circles indicate the wet deposition sites used in the analyses for the period 1990–2010, light circles indicate the extra sites used in the wet deposition analyses for 2000–2010 and the yellow triangles indicate the sites used for the evaluation of atmospheric concentrations. The black rectangle shows the domain used for the model simulations (for all models except CMAQ).**





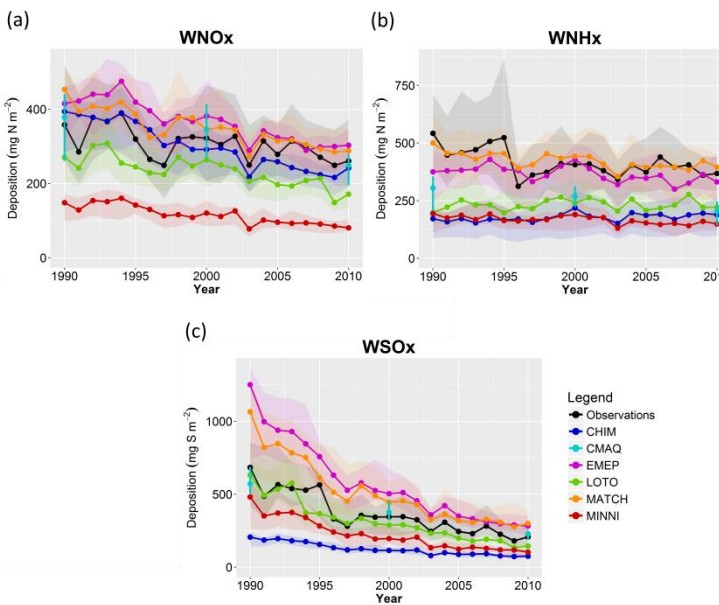

**Figure 2: Time series of observed and modelled annual wet deposition of a) WNOx, b) WNHx and c) WSOx. Points represent the annual median value for all measurement sites with a complete 21 year time series and the shading (or error bars) represents the interquartile range. The number of sites used for WNOx, WNHx and WSOx are 26, 21 and 20, respectively. Note: each plot has a different y-axis scale.**

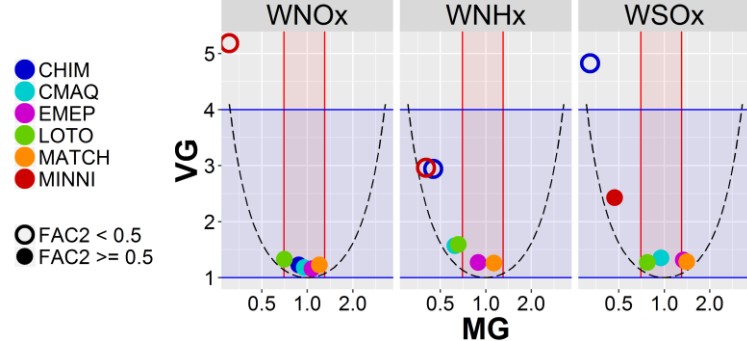

**Figure 3: Performance evaluation (geometric variance: VG vs. geometric mean bias: MG) of WNOx, WNHx and WSOx estimated by the six models that simulated the individual years 1990, 2000 and 2010. Shaded areas and filled symbols correspond to the acceptance criteria of Chang and Hanna (2004) (blue for VG, red for MG, filled circles for FAC2). Parabolic dashed lines indicate the theoretical minimum VG for a given value of MG.**





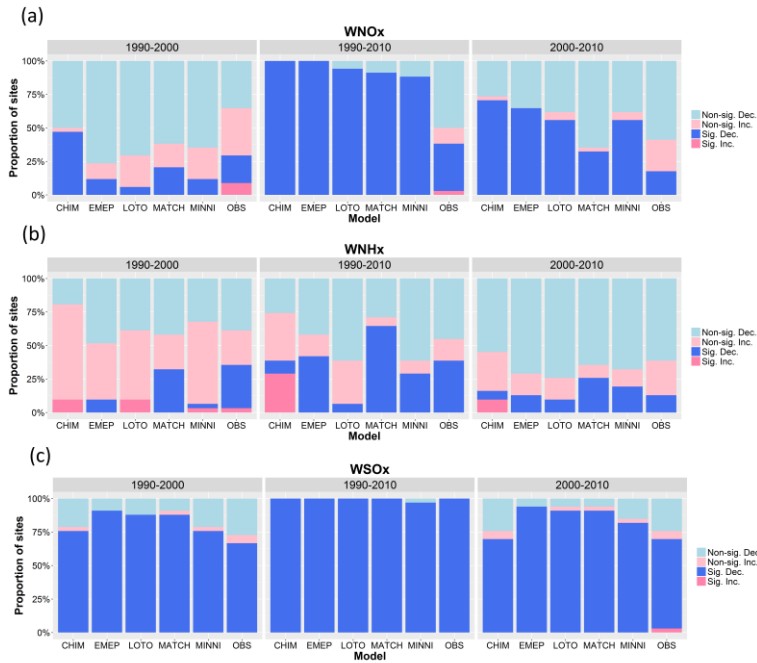

**Figure 4: Proportion of measurement sites with increasing (pink) and decreasing (blue) trends (and whether they are significant (dark colour) or not significant (light colour)) for the observations and model estimates for the three wet deposition components a) WNOx, b) WNHx and c) WSOx and the three time periods 1990–2000, 2000–2010 and 1990–2010 (left, middle and right columns).**





**Figure 5: Maps of modelled (coloured field) and observed (circles) trends in WNOx for the periods 1990–2000 and 2000–2010.**



**Figure 6: Maps of modelled (coloured field) and observed (circles) trends in WNHx for the periods 1990–2000 and 2000–2010.**





**Figure 7: Maps of modelled (coloured field) and observed (circles) trends in WSOx for the periods 1990–2000 and 2000–2010.**





**Figure 8: Tukey-style box plots of observed and modelled absolute (top row) and relative (middle row) trends for WNOx, WNHx, WSOx for the two periods 1990–2000 and 2000–2010 using the same set of sites for each period. The trends for all available sites for the second period (2000–2010 extra sites) are also shown in the plots of absolute trends (white boxes). Red and blue dotted lines in the plots of relative trends show the relative trends in total domain emissions for both periods. The bottom row shows the relative trends for the model simulations with constant emissions. All trends are shown, both significant and non-significant.**




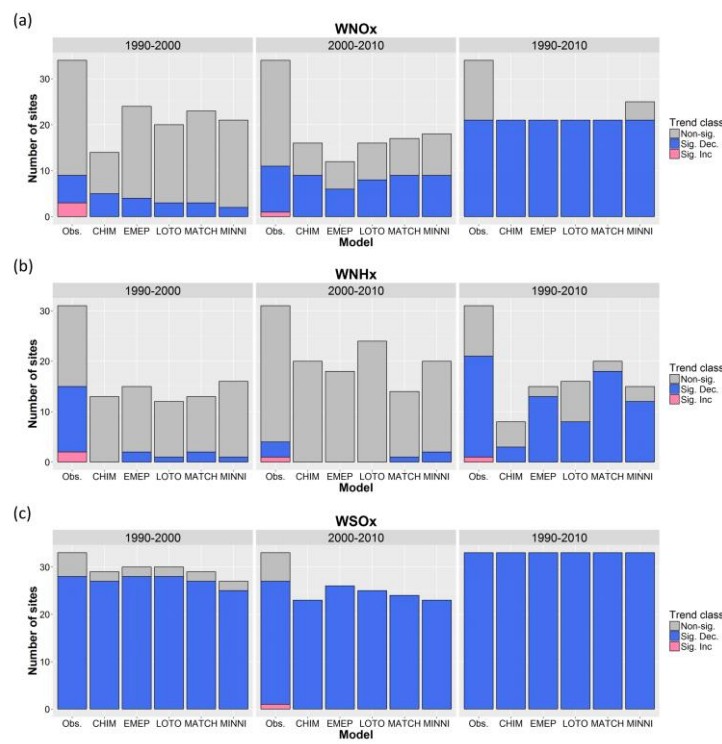

**Figure 9: Number of measurement sites with non-significant, significant increasing and significant decreasing observed trends and the number of sites for which the models classified the trends correctly. Plots are shown for the three wet deposition components a) WNOx, b) WNHx and c) WSOx and the three time periods 1990–2000, 2000–2010 and 1990–2010 (left, middle and right plots).**

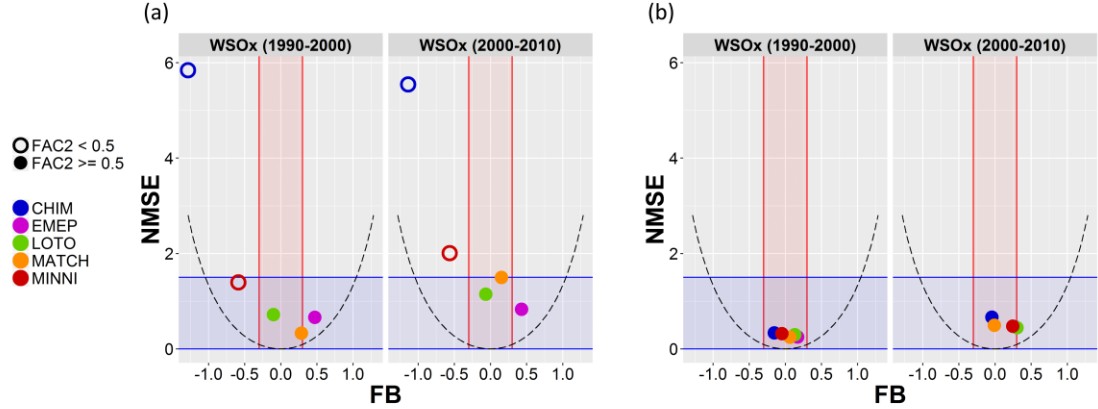

**Figure 10: Performance evaluation (Normalised mean square error: NMSE vs. fractional bias: FB) of the a) absolute and b) relative WSOx trends estimated by the five models for the two periods 1990–2000 and 2000–2010. Shaded areas and filled symbols correspond to the acceptance criteria of Chang and Hanna (2004) (blue for NMSE, red for FB, filled circles for FAC2). Parabolic dashed lines indicate the theoretical minimum NMSE for a given value of FB.**





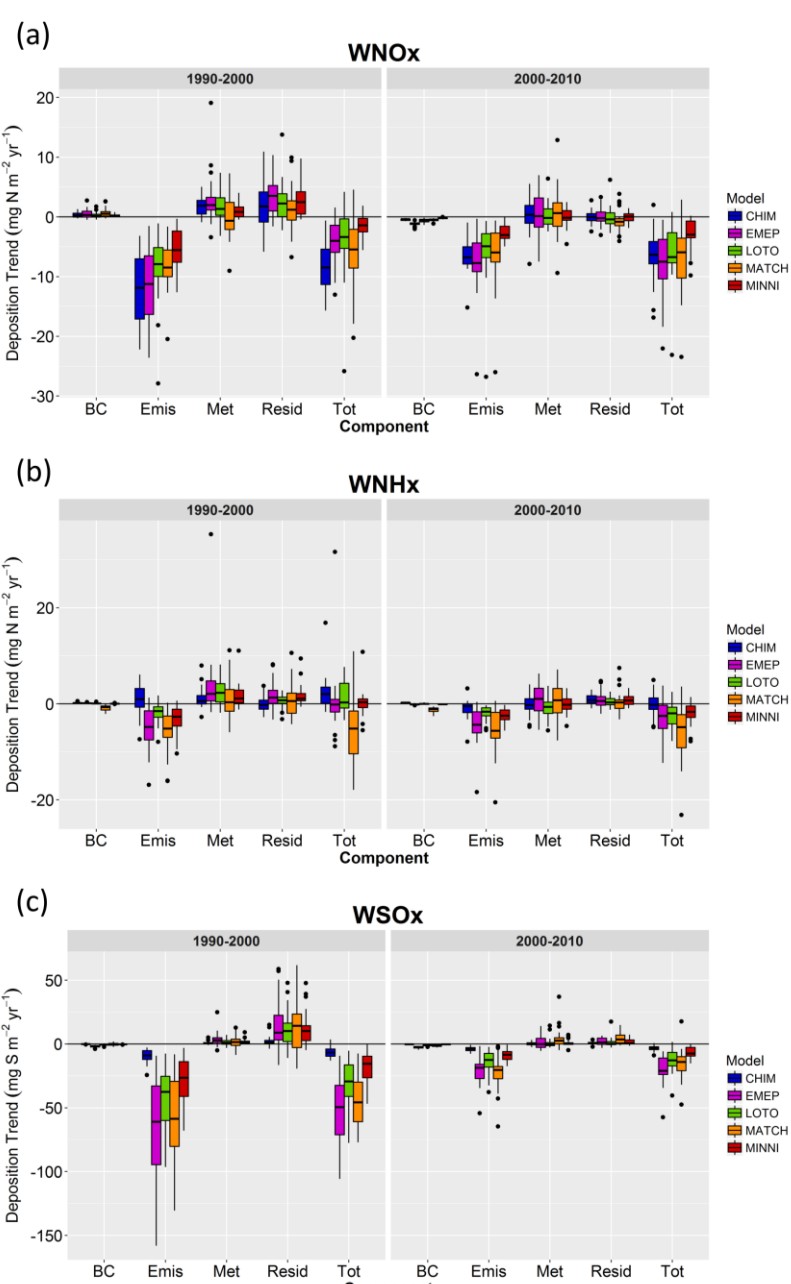

**Figure 11: Tukey-style box plots of the contributions of the different factors (BC: Boundary conditions; Emis: Emissions; Met: Meteorology and Resid: Residual interactions) to the trends (Tot) of a) WNOx, b) WNHx and c) WSOx at the sites with observations for the five models and two time periods.**





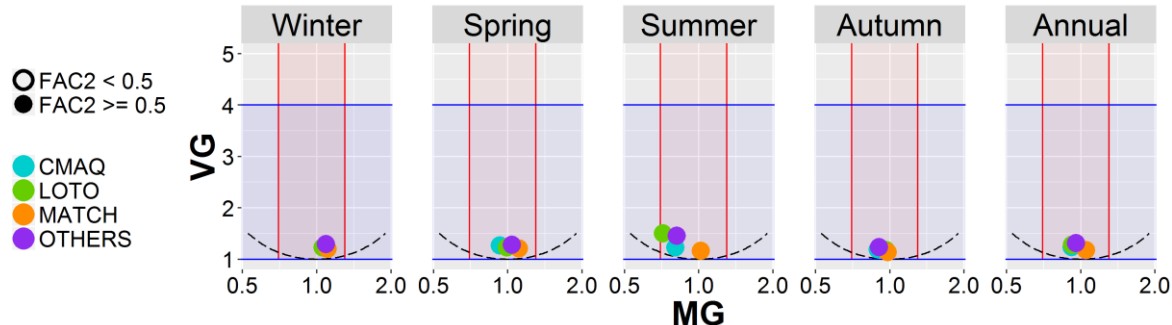

**Figure 12: Performance evaluation of the precipitation estimates for the meteorological data used in the simulations by CMAQ, LOTO, MATCH and the common meteorological data used in the other models (OTHERS). Shaded areas and filled symbols correspond to the acceptance criteria of Chang and Hanna (2004) (blue for VG, red for MG, filled circles for FAC2). Parabolic dashed lines indicate the theoretical minimum VG for a given value of MG (Chang and Hanna, 2004).**

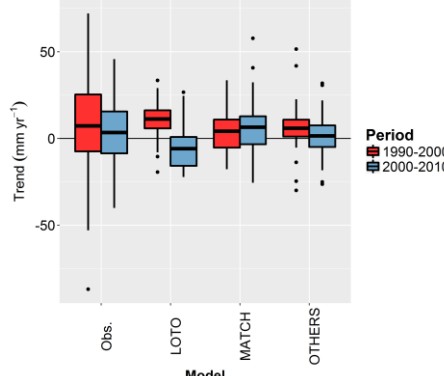

**Figure 13: Tukey-style box plots of observed and modelled trends in precipitation at the wet deposition sites for the two periods 1990–2000 and 2000–2010.**

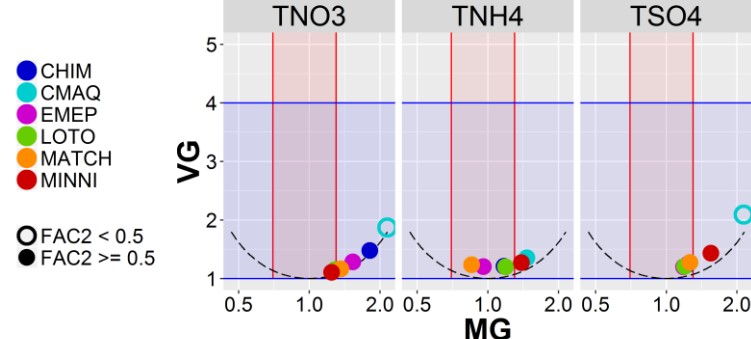

**Figure 14: Performance evaluation of the atmospheric concentrations of TNO3, TNH4 and TSO4 estimated by the six models that simulated the individual years 1990, 2000 and 2010. Shaded areas and filled symbols correspond to the acceptance criteria of Chang and Hanna (2004) (blue for VG, red for MG, filled circles for FAC2). Parabolic dashed lines indicate the theoretical minimum VG for a given value of MG.**





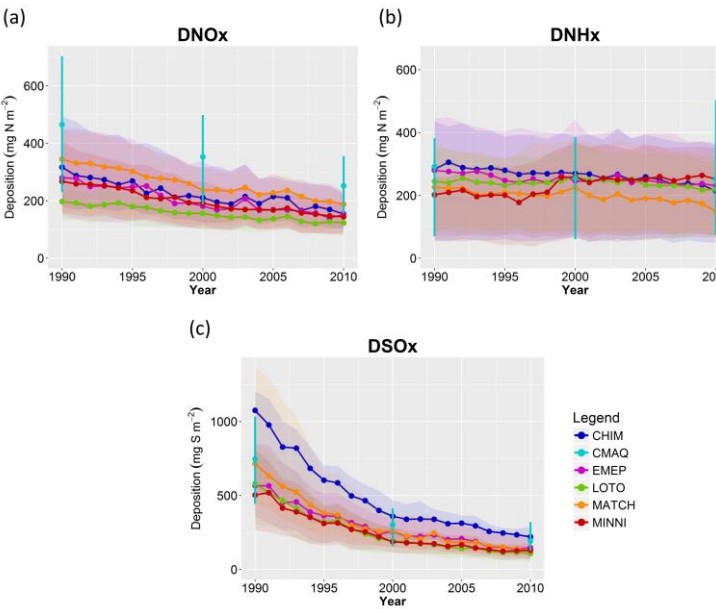

**Figure 15: Time series of modelled dry deposition of a) oxidised N (DNOx), b) reduced N (DNHx) and c) sulfur (DSOx). Points represent the median value for all measurement sites and the shading (or error bars) represents the interquartile range. Note: each plot has a different y-axis scale.**

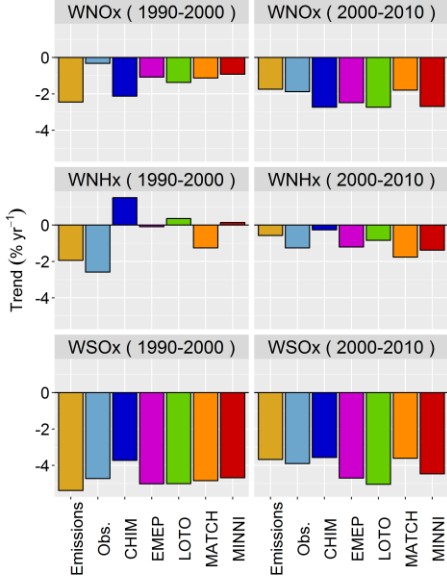

**Figure 16: A comparison of the relative trends of total domain emissions of precursor species (NOx for WNOx, NH₃ for WNHx and SOx for WSOx) and the median observed and modelled trends of WNOx, WNHx and WSOx at the measurement locations for the two eleven year time periods.**



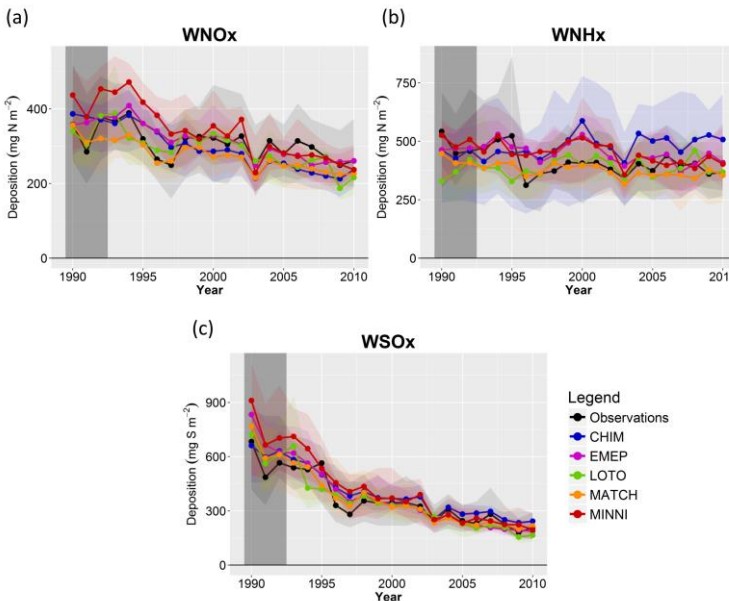

**Figure 17: Time series of observed and bias-corrected modelled wet deposition of WNOx, WNHx and WSOx. Points represent the median value for all measurement sites and the shading (or error bars) represents the interquartile range. The shaded period at the start of the time series represents the time period used to calculate the bias correction. Note: each plot has a different y-axis scale.**