# Peer review of "S1 Analysis of the errors in wet deposition due to errors in the primary particulate matter emissions"

_Atmospheric Chemistry and Physics, 2018_

## Referee Comment (RC1) · Anonymous Referee #1 · 2 Oct 2018

This paper presents an evaluation of modeled trends in wet deposition of sulfur and nitrogen compared to observations from the EMEP network for the periods 1990-2000 and 2000-2010. The paper provides a very detailed analysis of the trends, including examining factors contributing to model performance. Overall, the paper is well written, but in some sections becomes a bit if a recitation of statistics with little analysis. Section 3.6 is probably one of the more important sections, yet it is one of the shortest. Understanding why the observed trends are (or aren't) reproduced by the models is important. Page numbers or continuous line numbering would have been helpful.

[Figure]

Specific comments are as follows:

Page 3: What is the difference between the Collette et al. (2016) work and the Torseth et al. (2012) analysis?

Page 3, line 5: Consider a comma after "periods"

Page 3, line 6: Consider a comma before "but"

Page 3 - 4: There are several multi-model studies that are cited. It is impractical to provide the list of models and citations in this paper. It would be helpful to know, though, if the models used in the present study were included in those studies as well.

Page 6, line 10: If the other models were run with a lat-long grid, why wasn't CMAQ?

Page 7: Did any of the models include the bidirectional flux of NH3? This is noted in Table S2, but not discussed in the text. What is the impact on the model results of not considering this?

Page 7, line 7: Organic species were included in the modeled estimates of wet deposition. Are they included in the measurements? What about NO, NO2 and N2O5?

Page 7, line 13: Doesn't the CMAQ model provide information to distinguish sea-salt sulfate?

Page 7, line 20: consider rewording "network data of"

Page 8, line 13: Note that these criteria were developed for atmospheric concentrations and not deposition values.

Page 8, line 17-18: Clarify what the observed and modelled trends are for on line 17 and what trends on line 18 are more difficult to evaluate compared to annual wet deposition.

Page 8, lines 20-23: suggest splitting the sentence at "then" on line 20.

Page 8, line 21: "were" should be "was" as it refers to magnitude

Page 9, line 2: How were the tau values determined?

Page 9, line 24: Does "European" start a new paragraph?

Page 9, lines 25-30: Specific information is given from Sutton et al. (2003) about why NH3 emissions decreased but he same level of detail is not provided for other species.

Page 10, line 15: Consider listing the meteorological models

Page 10, line 19: Consider specifying "meteorological models" rather than just models.

Page 10, line 25: It seems odd that one WRF run (used for CMAQ) would have such very different precipitation compared to the other WRF runs. What was different about the WRF runs? It might be helpful to have a table in the supplemental that provides details on the meteorological models.

Page15, section 23.5: This section seems to repeat information that was presented earlier.

Page 21, lines 30-33: Do these studies use different versions of the EMP model? Please indicate what versions were used.

Page 22, line 13: Why is the trend for observed WNOx for 1990-200 in Figure 16 so different than the emissions trend? Is this realistic?

Page 24, line18-19: what would be the effect on mass conservation of doing a bias correction?

Figures 4 and 9: The legend text is too small.

Figure 12: Are these period (i.e. seasonal) totals values?

Table S2: - Consider adding a table with specifics of the met model runs - Are the vertical layers for the CTM or the met model? - For Chimere, CMAQ, and MINNI, give an approximate value for the 1st model layer. - CMAQ description is incomplete and incorrect. No citation is given for the dry deposition of gases. CMAQ does include a

bidirectional NH3 model (but maybe it wasn't used). Wesely (1989) is not the correct reference for the stomatal resistance. This is calculated in the Pleim-Xu land surface model and is described in papers by Pleim and Xu.

---

## Referee Comment (RC2) · Anonymous Referee #2 · 2 Oct 2018

The manuscript presents the comparison of the trends of sulphur and nitrogen wet deposition from 4 CTMs to the observations of EMEP Network. The results of the study are of interest because such models are commonly applied to model the impacts of future emission scenarios, creating a need for the knowledge of their reliability at reproducing the trends observed due to past emission changes. The paper is written in good English. The methods are well described and seem sound. However, the major shortcoming of the paper is the tendency to flood the reader with too much minor detail, making the reading tedious. I think the main messages could be delivered better by

substantially cutting the length of the paper and leaving majority of the specific details about the skill of the individual models in tables instead of the main text, especially as the authors state that providing in depth analysis of the models' performance or inter-model differences it is out of the scope of their study.

Specific comments:

1. Page 2, line 23. Why specifically semi-natural vegetation?

2. The analysis of previously published trends in observational data is currently cut to two parts (before and after the CTM results), making the structure of the introduction confusing and prone to repetition. This text includes too many details and all the specific numbers from all these studies would be far better visible and understandable if presented as a table.

3. The overview of previous model-measurement comparisons could also be substantially shortened, as naming the specific models participating in those studies does not provide extra information, with the possible exception of if these are the same models as used in this study and this information is later used for discussion. I would suggest to try to compress this information into a few sentences per species, giving the general view whether the previous studies have shown any consistent under- or overestimation of its wet deposition. Or, if needed, including a supplementary table with the detailed numbers from these studies.

4. Please provide the reason why the 21-year period was divided to two 11-year sub-periods.

5. Were the NOx and NH3 emissions from wildfires included? How about SO2 from volcanoes?

6. Could the CMAQ results be corrected for sea-salt sulphate (for instance using Na concentration in similar manner to how observations are corrected)?

7. The description of emission changes could be shortened, for instance combining

what happened to shipping emissions of both NOx and SOx into a single sentence and reducing the listings of specific countries and values.

8. Spatial distributions are compared for 3 years (1990, 2000 and 2010) - are the differences in the patterns between these specific years representative of the overall trends?

9. The paper could be shortened by skipping naming the models which simulated the largest and smallest results in majority of occasions apart from those few where the reason for the outlying model result is given.

10. Page 11, lines 28-31: If the emission data was given at 5-year interval and interpolated between the given years, the models cannot be expected to perfectly reproduce year-to-year variability which might result from instant changes in some emission sources due to closing of some facilities or implementation of emission control measures.

11. Page 19, lines 27 – page 20, line 1 - "the net effect of these uncertainties is not expected to be a large systematic under- or overestimation of wet deposition." Due to the highly soluble nature of the compounds discussed here relatively little precipitation is needed for almost complete removal of them from the below-cloud column, leading to strong non-linearity of the wet deposition process. Thus, errors in modelled rain frequency might be more relevant for modelling the wet deposition than the annual precipitation amount and too frequent light rains instead of a few strong ones can for instance easily lead to positive bias in wet deposition.

Technical corrections:

1. Table 1. The optimal value of the geometric variance should be 1.

2. Figure 4 – Please correct the caption - the three time periods 1990–2000, 2000–2010 and 1990–2010 (left, middle and right columns) seem to be actually 1990–2000, 1990–2010, and 2000–2010

---

## Author Comment (AC2) · 28 Nov 2018

**Response to Referees Comments**

*AR: Authors' response*

**Anonymous Referee #1**

This paper presents an evaluation of modeled trends in wet deposition of sulfur and nitrogen compared to observations from the EMEP network for the periods 1990-2000 and 2000-2010. The paper provides a very detailed analysis of the trends, including examining factors contributing to model performance. Overall, the paper is well written, but in some sections becomes a bit if a recitation of statistics with little analysis. Section 3.6 is probably one of the more important sections, yet it is one of the shortest. Understanding why the observed trends are (or aren't) reproduced by the models is important. Page numbers or continuous line numbering would have been helpful.

*AR: We thank the referee for their constructive comments. In the revised manuscript we have shortened the analysis by removing references to the performance metrics of individual models except where we want to highlight the performance of a particular model, for example a model that gives a large bias. We have also expanded Section 3.6 (Trend attribution analysis, now Section 3.5) by including an analysis of the spatial distributions of the factors influencing the trends. This new analysis suggests that the influence of changing meteorology on the wet deposition trends is mostly due to changing precipitation patterns during the two periods and that the "Residual" component is also driven by changes in precipitation. This gives strength to our suggestion that changes in precipitation partially offset the decreasing trends due to emission reductions during the first period but not the second, at the measurement sites.*

**Specific comments:**

Page 3: What is the difference between the Collette et al. (2016) work and the Torseth et al. (2012) analysis?

*AR: The main difference between these two studies is the time periods they cover. Torseth et al. covers the period 1980–2009 whereas Collette et al. covers the period 1990–2012. In fact the study by Colette et al. was designed as an extension to that of Torseth et al. with updated methodologies and site selection, which were agreed on during meetings of the Task Force on Measurements and Modelling and a dedicated workshop. Collette et al. also contains additional information, such as the modelled air quality trends. In the revised manuscript we have reduced the size of this section and now only include a summary of the trends estimated from observations without listing the trends from each study.*

Page 3, line 5: Consider a comma after "periods"

*AR: A comma has been added*

Page 3, line 6: Consider a comma before "but"

*AR: A comma has been added*

Page 3 - 4: There are several multi-model studies that are cited. It is impractical to provide the list of models and citations in this paper. It would be helpful to know, though, if the models used in the present study were included in those studies as well.

*AR: The results of the multi-model studies have been summarised to remove the detail and highlight the variability of model performance for wet deposition estimates. Previous results of models used in this study are discussed in the Discussion section.*

Page 6, line 10: If the other models were run with a lat-long grid, why wasn't CMAQ?

*AR: For the CMAQ model the horizontal grid coordinate system is the same as the other models (i.e. latitude-longitude), however the CMAQ model uses Lambert conformal Conic map projection in its native state with 25 km resolution. To be comparable with the other models, the CMAQ output was interpolated to the common domain used by the other models. Also note that the common domain consists of a regular latitude–longitude grid with increments of 0.25° and 0.4° in the latitude and longitude, respectively, which is about 25 km × 25 km at European latitudes. This means that both grids are comparable.*

Page 7: Did any of the models include the bidirectional flux of NH3? This is noted in Table S2, but not discussed in the text. What is the impact on the model results of not considering this?

*AR: Only one model, LOTO-EUROS, included bidirectional fluxes of NH3. This model includes compensation points for stomata and leaf, soil and water surfaces (although the compensation point for soil surfaces is currently set to zero). Wichink Kruit et al. (2012) showed that the inclusion of compensation points in the LOTOS-EUROS model decreased annual $NH_3$ dry deposition, especially in ammonia source areas, leading to an increase in the atmospheric lifetime of $NH_3$ and an increase in WNHx over most of the continent. However, the relative increases in WNHx were very small over land areas and were much smaller than the inter-model differences found in our study. EMEP has a simplified approach with no dry deposition of $NH_3$ to growing crops, which also increases $NH_3$ concentrations slightly. If compensation points were included in the other models then this would be expected to increase the estimates of WNHx slightly although it would not be enough to correct the negative biases found for some of the models. We have added these comments to section 4.2 in the revised manuscript.*

*Ref.: Wichink Kruit, R. J., et al. "Modeling the distribution of ammonia across Europe including bi-directional surface–atmosphere exchange." Biogeosciences 9.12 (2012): 5261-5277.*

Page 7, line 7: Organic species were included in the modeled estimates of wet deposition. Are they included in the measurements? What about NO, NO2 and N2O5?

*AR: Organic species are not included in the measurements, although they should be considering that they are estimated to contribute to around 25% of N wet deposition in Europe (Cornell, 2011). Taking CHIMERE as an example, although the model includes organic species in the wet deposition estimates the actual contribution is zero for these simulations. It is expected that the contribution from organic species (if they include them) in the wet deposition output of the other models is also zero or negligible. By contrast, CHIMERE estimates that organic species contribute up to 13% of the grid cell dry deposition of oxidised nitrogen, which is clearly not negligible although we have not evaluated dry deposition in this study. NO and $NO_2$ are*

*relatively insoluble in water compared to other gases such as $NH_3$, $HNO_3$ and $SO_2$ and so are not expected to make a large contribution to the measured N wet deposition. $N_2O_5$, however is highly soluble but atmospheric concentrations are generally quite low and so its concentration to wet deposition is also expected to be small.*

*Ref.: Cornell, Sarah. "Atmospheric nitrogen deposition: revisiting the question of the invisible organic fraction." Procedia Environmental Sciences 6 (2011): 96-103.*

Page 7, line 13: Doesn't the CMAQ model provide information to distinguish sea-salt sulfate?

*AR: CMAQ provides sulfate in three modes (Aiken, accumulation and coarse) without distinguishing their source. In this version of the model, the coarse sulfate is from sea salt emissions. In this revised analysis we removed the coarse sulfate from the CMAQ estimates of total sulphate concentrations, as now described at the end of Section 2.1.*

Page 7, line 20: consider rewording "network data of"

*AR: The sentence has been rewritten as "For the evaluation of modelled atmospheric concentration estimates, the EMEP network data of mean annual concentrations of total nitrate, ammonium and sulfate (non-sea-salt component) were used." [Page 6, lines 16-18]*

Page 8, line 13: Note that these criteria were developed for atmospheric concentrations and not deposition values.

*AR: This is a very valid point and the following disclaimer has been added to the manuscript "It should be noted, however, that these criteria were developed for evaluating the atmospheric concentrations estimated by air quality models using specially designed model evaluation field experiments. They may not, therefore, be an appropriate tool for evaluating operational wet deposition estimates using monitoring data and can only be used as an indicator of model acceptability." [Page 7, lines 13-16]*

Page 8, line 17-18: Clarify what the observed and modelled trends are for on line 17 and what trends on line 18 are more difficult to evaluate compared to annual wet deposition.

*AR: Line 17 refers to the observed and modelled wet deposition trends and line 18 refers to the evaluation of the modelled wet deposition trends. However, the statistical evaluation of the wet deposition trends has been removed in the revised manuscript since it was considered confusing and did not contribute much additional information to the analyses.*

Page 8, lines 20-23: suggest splitting the sentence at "then" on line 20.

*AR: The sentence has been modified and split into two sentences*

Page 8, line 21: "were" should be "was" as it refers to magnitude

*AR: OK, the change has been made*

Page 9, line 2: How were the tau values determined?

*AR: This is described in the subsequent lines: "..approximated as the difference in wet deposition over the eleven year period for simulations where the other factors are kept constant, divided by ten (to obtain the mean annual trend). For example, the change in wet deposition over the period 1990–2000 due to changes in emissions can be calculated from two simulations with emissions for 1990 and 2000, both with meteorology and boundary conditions for 2000."*

Page 9, line 24: Does "European" start a new paragraph?

*AR: Yes it does. The preceding paragraph is on the NOx emission trends and the new paragraph is on the NH$_3$ emission trends*

Page 9, lines 25-30: Specific information is given from Sutton et al. (2003) about why NH3 emissions decreased but he same level of detail is not provided for other species.

*AR: This decrease was specifically highlighted since it was the result of political change and not a result of the implementation of control measures to reduce emissions mentioned in the introduction*

Page 10, line 15: Consider listing the meteorological models

*AR: We agree. The model names have been included in the revised manuscript*

Page 10, line 19: Consider specifying "meteorological models" rather than just models.

*AR: We agree. We have made the recommended change*

Page 10, line 25: It seems odd that one WRF run (used for CMAQ) would have such very different precipitation compared to the other WRF runs. What was different about the WRF runs? It might be helpful to have a table in the supplemental that provides details on the meteorological models.

*AR: The main difference between the two WRF simulations is that the simulation used as the common meteorological driver used nudging whereas the WRF simulation used for the CMAQ simulations was free-running (only forced at the domain boundaries). This difference along with the different grid spacing could lead to the discrepancy in precipitation. The use or not of nudging has been added to Table S1. The specifics of the met model runs are presented in Table 4 of Colette et al. (2017a) and we considered that it was not efficient reproducing this information in the manuscript.*

Page15, section 3.5: This section seems to repeat information that was presented earlier.

*AR: This section presents a statistical evaluation of the modelled trends at the measurement sites by comparing the modelled and observed trends (and their significance) at each site. This is different to the results presented in the preceding section which compares the distributions of the modelled and observed trends (i.e. not a direct comparison of the modelled and observed trends at each site) in order to show the bias of modelled trends, on average. However, section 3.5 was considered confusing and did not contribute much additional information to the analyses and has been removed in the revised manuscript.*

Page 21, lines 30-33: Do these studies use different versions of the EMEP model? Please indicate what versions were used.

*AR: Yes they did use different versions. This has been clarified in the manuscript*

Page 22, line 13: Why is the trend for observed WNOx for 1990-200 in Figure 16 so different than the emissions trend? Is this realistic?

*AR: Yes, we believe that it is realistic. This difference is due to low significance of the observed WNOx trends for this period. Increasing trends were observed at 15 of the 34 sites but only 3 of these were significant (see Fig. 4). Decreasing trends were observed at 19 sites but only 6 were significant. The fact that increasing trends were observed at 44% of the sites and decreasing trends at 56% of the sites leads to a median trend close to zero. If only the significant trends are used (26% of sites), the median trend is -3.3% per year, which is larger than the emission trend. However, the sites with the most significant trends are most likely to be located in the regions with the largest emission reductions and, therefore, are not representative of the model domain.*

Page 24, line18-19: what would be the effect on mass conservation of doing a bias correction?

*AR: The bias correction will invalidate any assumptions of mass conversation since the correction only applies to the simulated wet deposition, leaving other components (e.g. atmospheric concentrations) unchanged. The bias correction is proposed as a post-processing step to provide more accurate estimates of future wet deposition. If mass conservation is required for these estimates, however, the bias correction should not be applied.*

Figures 4 and 9: The legend text is too small.

*AR: We have increased the legend text size of Figure 4 in the revised manuscript. Figure 9 is not in the revised manuscript*

Figure 12: Are these period (i.e. seasonal) totals values?

*AR: Yes they are. This has been clarified in the caption*

Table S2: - Consider adding a table with specifics of the met model runs - Are the vertical layers for the CTM or the met model? - For Chimere, CMAQ, and MINNI, give an approximate value for the 1st model layer. - CMAQ description is incomplete and incorrect. No citation is given for the dry deposition of gases. CMAQ does include a bidirectional NH3 model (but maybe it wasn't used). Wesely (1989) is not the correct reference for the stomatal resistance. This is calculated in the Pleim-Xu land surface model and is described in papers by Pleim and Xu.

*AR: The specifics of the met model runs are presented in Table 4 of Colette et al. (2017a) and we considered that it was not efficient reproducing this information in the manuscript. However, we now refer to this table in the text so that the reader can easily access this information if they require. The vertical layers shown are for the CTM. This has been clarified in Table S1 of the revised Supplementary Material and we have also included an estimate of the depth of the first layer used in Chimere, CMAQ and MINNI. The information regarding the parameterisation of CMAQ has also been corrected.*

---

## Author Comment (AC3) · 28 Nov 2018

**Response to Referees Comments**

*AR: Authors' response*

**Anonymous Referee #2**

The manuscript presents the comparison of the trends of sulphur and nitrogen wet deposition from 4 CTMs to the observations of EMEP Network. The results of the study are of interest because such models are commonly applied to model the impacts of future emission scenarios, creating a need for the knowledge of their reliability at reproducing the trends observed due to past emission changes. The paper is written in good English. The methods are well described and seem sound. However, the major shortcoming of the paper is the tendency to flood the reader with too much minor detail, making the reading tedious. I think the main messages could be delivered better by substantially cutting the length of the paper and leaving majority of the specific details about the skill of the individual models in tables instead of the main text, especially as the authors state that providing in depth analysis of the models' performance or inter-model differences it is out of the scope of their study.

*AR: We thank the referee for their constructive comments. In the revised manuscript we have shortened the analysis by removing references to the performance metrics of individual models except where we want to highlight the performance of a particular model, for example a model that gives a large bias.*

**Specific comments:**

1. Page 2, line 23. Why specifically semi-natural vegetation?

*AR: This term is frequently used to refer to non-managed or extensively-managed ecosystems (e.g. woodland, moorland, meadows, mountain habitats etc.) in recognition that there are very few ecosystems in Europe that have not been directly influenced by human activity. To avoid confusion, we have changed this to "natural and semi-natural" [Page 2, line 22]*

2. The analysis of previously published trends in observational data is currently cut to two parts (before and after the CTM results), making the structure of the introduction confusing and prone to repetition. This text includes too many details and all the specific numbers from all these studies would be far better visible and understandable if presented as a table.

*AR: In the revised manuscript we have unified the two parts and reduced the size of this section, including only a summary of the (range of) trends estimated from observations without listing the trends from each study.*

3. The overview of previous model-measurement comparisons could also be substantially shortened, as naming the specific models participating in those studies does not provide extra information, with the possible exception of if these are the same models as used in this study and this information is later used for discussion. I would suggest to try to compress this information into a few sentences per species, giving the general view whether the previous studies have shown any consistent under- or overestimation of its wet deposition. Or, if needed, including a supplementary table with the detailed numbers from these studies.

*AR: The results of the previous model-measurement comparisons have been summarised to remove the detail and highlight the variability of model performance for wet deposition estimates as well as give an indication of tendencies of models to under- or overestimate wet deposition (median values of normalised bias for each species)*

4. Please provide the reason why the 21-year period was divided to two 11-year subperiods.

*AR: This was done to be able to calculate trends for two ten-year trends (1990-2000 and 2000-2010). This has been clarified in the revised manuscript. In addition, the emission trends over the 1990s are larger than over the 2000s and so deposition trends may be non-linear for over the full simulation period.*

5. Were the NOx and NH3 emissions from wildfires included? How about SO2 from volcanoes?

*AR: Emissions from wildfires were not included in any model and volcanic emissions of $SO_2$ were only included in the simulations by EMEP and MATCH. The following sentence has been added to clarify this "Emission from wildfires were not included and SO2 emissions from volcanoes were only included in the EMEP (Etna and Stromboli) and MATCH models." [Page 5, lines 29-31]*

6. Could the CMAQ results be corrected for sea-salt sulphate (for instance using Na concentration in similar manner to how observations are corrected)?

*AR: In CMAQ, marine sulfate is emitted directly in the coarse fraction, so considering only $PM_{2.5}$ sulfate will give the total anthropogenic sulfate. The evaluation of atmospheric concentrations has been modified so that total sulfate concentrations for CMAQ are calculated using the $PM_{2.5}$ sulphate concentrations, not the $PM_{10}$ concentrations as done for the other models. Unfortunately it is not possible to separate out the contribution of sea-salt sulfate to WSOx in a similar way because the modelled contributions from $PM_{2.5}$ and $PM_{10}$ are not provided separately. However, the non-corrected observations are available and so now the WSOx estimated by CMAQ are evaluated using the non-corrected data. The methods and results have been updated accordingly.*

7. The description of emission changes could be shortened, for instance combining what happened to shipping emissions of both NOx and SOx into a single sentence and reducing the listings of specific countries and values.

*AR: This section has been shortened by removing specific details of the emission trends and combining the description of shipping emission trends for NOx and SOx, as suggested by the referee*

8. Spatial distributions are compared for 3 years (1990, 2000 and 2010) - are the differences in the patterns between these specific years representative of the overall trends?

*AR: These years were not chosen to be representative although they do show fairly representative changes for situations with large trends (e.g. estimates of WSOx). These three years were chosen simply because they were the years that were simulated by all models*

9. The paper could be shortened by skipping naming the models which simulated the largest and smallest results in majority of occasions apart from those few where the reason for the outlying model result is given.

*AR: As mentioned above, in the revised manuscript we have shortened the analysis by removing references to the performance metrics of individual models except where we want to highlight the performance of a particular model, for example a model that gives a large bias.*

10. Page 11, lines 28-31: If the emission data was given at 5-year interval and interpolated between the given years, the models cannot be expected to perfectly reproduce year-to-year variability which might result from instant changes in some emission sources due to closing of some facilities or implementation of emission control measures.

*AR: This is true and it is one of the limitations of this emission dataset. However, with regards to the modelled deposition estimates, the inter-annual variability due to the meteorology is expected to be larger than that due to the emissions although, this may not be the case at certain locations due to the issues mentioned by the referee*

11. Page 19, lines 27 – page 20, line 1 - "the net effect of these uncertainties is not expected to be a large systematic under- or overestimation of wet deposition." Due to the highly soluble nature of the compounds discussed here relatively little precipitation is needed for almost complete removal of them from the below-cloud column, leading to strong non-linearity of the wet deposition process. Thus, errors in modelled rain frequency might be more relevant for modelling the wet deposition than the annual precipitation amount and too frequent light rains instead of a few strong ones can for instance easily lead to positive bias in wet deposition.

*AR: This is a good point and one that we have identified during our analysis as a subject of future evaluation studies. Doing this properly would require an analysis of the hourly measured and modelled precipitation and wet deposition (where available) for each model. This analysis is out of the scope of the current evaluation, which focuses on accumulated annual wet deposition and its trends over a twenty year period. We have highlighted this subject in the revised manuscript as a focus for future studies with the following sentences "In addition to the uncertainties in annual accumulated precipitation, the departure of the hourly, daily and monthly modelled precipitation from the observed values could lead to large errors in the modelled wet deposition for some models in some locations. The assessment of this effect would require an analysis of the hourly observed and modelled precipitation, atmospheric concentrations and wet deposition and should be considered for future analyses" [Page 17, lines 4-8]*

**Technical corrections:**
1. Table 1. The optimal value of the geometric variance should be 1.

*AR: This has been corrected in the revised manuscript*

2. Figure 4 – Please correct the caption - the three time periods 1990–2000, 2000–2010 and 1990–2010 (left, middle and right columns) seem to be actually 1990–2000, 1990–2010, and 2000–2010

*AR: The columns in the revised figure are now in the same order as the labels in the caption with the two 11 year time periods followed by the 21 year time period*

---

## Author Comment (AC4) · 28 Nov 2018

**An evaluation of European nitrogen and sulfur wet deposition and their trends estimated by six chemistry transport models for the period 1990–2010**

Mark R. Theobald1, Marta G. Vivanco1, Wenche Aas2, Camilla Andersson3, Giancarlo Ciarelli4, Florian Couvidat5, Kees Cuvelier6, Astrid Manders7, Mihaela Mircea8, Maria-Teresa Pay9, Svetlana Tsyro10, Mario Adani8, Robert Bergström3,11, Bertrand Bessagnet5, Gino Briganti8, Andrea Cappelletti8, Massimo D'Isidoro8, Hilde Fagerli10, Kathleen Mar12, Noelia Otero12, Valentin Raffort13, Yelva Roustan13, Martijn Schaap7,14, Peter Wind10,15 and Augustin Colette5

1Atmospheric Pollution Unit, CIEMAT, Avda. Complutense, 40, 28040 Madrid, Spain

- 2Norwegian Institute for Air Research (NILU), Box 100, 2027 Kjeller, Norway
   3Swedish Meteorological and Hydrological Institute, 60176 Norrköping, Sweden
   4Laboratoire Inter-Universitaire des Systèmes Atmosphériques (LISA), UMR CNRS 7583, Université Paris Est Créteil et Université Paris Diderot, Institut Pierre Simon Laplace, Créteil, France
   5National Institute for Industrial Environment and Risks (INERIS), Parc Technologique ALATA, F-60550 Verneuil-en Halatte, France
   6ex European Commission, Joint Research Centre (JRC), Ispra, Italy
  - 7Netherlands Organisation for applied scientific research (TNO), P.O. Box 80015, 3508 TA Utrecht, The Netherlands
     8Italian National Agency for New Technologies, Energy and Sustainable Economic Development (ENEA), Via Martiri di Monte Sole 4, 40129 Bologna, Italy
- 9Barcelona Supercomputing Center, Centro Nacional de Supercomputación, Jordi Girona, 29, 08034 Barcelona, Spain 10Climate Modelling and Air Pollution Division, Research and Development Department, Norwegian Meteorological Institute (MET Norway), Blindern, N-0313 Oslo, Norway 11Chalmers University of Technology, Gothenburg, SE-412 96, Sweden
  - 12Institute for Advanced Sustainability Studies (IASS), Postdam, Germany

[revised manuscript text omitted]

5

20

| Tabl | e 1: | The s | ix perfe | ormance | metrics | relatin | g mode   | l estima | ntes (M | i) to          | the observ | ed val | ues (C | );) used | to assess | model | performance. |
|------|------|-------|----------|---------|---------|---------|----------|----------|---------|----------------|------------|--------|--------|----------|-----------|-------|--------------|
|      |      |       |          |         |         |         | - |          |         | v · · · |            |        |        | P        |           |       |              |

| Performance metric                          | Definition                                                                                                                                 | Optimum value |
|---------------------------------------------|--------------------------------------------------------------------------------------------------------------------------------------------|---------------|
| Fraction of model estimates within a factor | $0.5 \le \frac{M_i}{2} \le 2.0$                                                                                                            | 1             |
| of two of the observations (FAC2)           | $O_i = O_i$                                                                                                                                | 1             |
| Fractional bias (FB)                        | $FB = \frac{2(\overline{M} - \overline{O})}{(\overline{M} + \overline{O})}$                                                                | 0             |
| Geometric mean bias (MG)                    | $MG = \exp(\overline{\ln M} - \overline{\ln O})$                                                                                           | 1             |
| Normalised mean square error (NMSE)         | $NMSE = \frac{\overline{(O-M)^2}}{\overline{O}\ \overline{M}}$                                                                             | 0             |
| Geometric variance (VG)                     | $VG = exp\left[(\ln O - \ln M)^2\right]$                                                                                                   | 1      |
| Pearson correlation coefficient (r)         | $r = \frac{1}{(n-1)} \sum_{i=1}^{n} \left( \frac{M_i - \overline{M}}{\sigma_M} \right) \left( \frac{O_i - \overline{O}}{\sigma_O} \right)$ | 1             |

---

## Author Comment (AC5) · 28 Nov 2018

**S1 Analysis of the errors in wet deposition due to errors in the primary particulate matter emissions**

Errors were found in the emissions of primary particulate matter for Russia and North African countries and shipping for the period 1991-1999. Unfortunately it was not possible to re-run the simulations since these errors were not detected until late in the data analysis. In order to estimate the impact on the wet deposition estimates, the CHIMERE model was used to simulate wet deposition using the incorrect and corrected emissions for 1998, the year for which the emission error is the largest. Figure S1 shows the relative errors in the model estimates of WNOx, WNHx and WSOx as a result of the errors in emissions. Errors in WNOx and WSOx were less than 0.5% in most of the domain with maximum errors of 0.95% and 1.5%, respectively. Errors in WNHx were also mostly below 0.5% but larger errors were estimated for about a quarter of the domain (mostly in the northeast), with a maximum error of 2.4%. These errors are small compared with the overall uncertainty of the model estimates and the uncertainty of the observations. Errors in annual deposition rates. From this analysis we conclude that the error in emissions is unlikely to affect the results and conclusions of the study significantly.

Figure S1: Maps of relative difference between the model estimates of WNOx, WNHx and WSOx for the simulations using the original (incorrect) emissions and those using the corrected emissions of primary PM in Russia, North Africa and maritime areas for 1998. Circles show the locations of the sites used to evaluate modelled deposition. Note: Positive values indicate that the emission error resulted in larger values of wet deposition and vice versa.

**S2 Table and Figures cited in the article**

 Table S1: Main features of the chemistry-transport models involved in the EURODELTA-Trends deposition modelling exercise (Adapted from Colette et al., 2017a).

| Model                                                     | CHIMERE (CHIM)                                        | СМАQ                                    | EMEP MSC-W (EMEP)                                   | LOTOS-EUROS
(LOTO)                    | МАТСН                                               | MINNI                                               |
|-----------------------------------------------------------|-------------------------------------------------------|-----------------------------------------|-----------------------------------------------------|------------------------------------------|-----------------------------------------------------|-----------------------------------------------------|
| Version / Date                                            | Modified CHIMERE2013                                  | V5.0.2                                  | rv4.7 spring 2015                                   | v1.10.005                                | VSOA April 2016                                     | V4.7                                                |
| Operator                                                  | INERIS                                                | BSC                                     | MET Norway                                          | TNO                                      | SMHI                                                | ENEA/Arianet S.r.l.                                 |
| Name and
resolution of the
meteorological
driver | WRF with nudging
(common driver)
0.44°   | WRF (no nudging)
25 km        | WRF with nudging
(common driver)
0.44° | RACMO2
0.22°                          | HIRLAM EURO4M
reanalysis
Approx. 22 km        | WRF with nudging
(common driver)
0.44° |
| Vertical layers of
CTM                          | 9 sigma                                               | 15 sigma                                | 20 sigma                                            | 5 (4 dynamic layers and a surface layer) | 39 hybrid levels of the meteorological model layers | 16 fixed terrain-following layers                   |
| Vertical extent of
CTM                   | 500 hPa                                               | 50 hPa                                  | 100 hPa                                             | 5000 m                                   | ca. 5000 m (4700–6000
m)                         | 10 000m                                             |
| Surface
concentration
height                 | 10 m (midpoint of first
model Jayer) | 20 m (midpoint of first
model Jayer) | Downscaled to 3 m                                   | Downscaled to 3 m                        | Downscaled to 3 m                                   | 20 m (midpoint of first
model Jayer)             |
| Land-use
database                                      | GLOBCOVER (24 classes)                                | Corine Land Cover 2006
(44 classes)  | CCE/SEI for Europe, elsewhere GLC2000               | Corine Land Cover 2000 (13 classes)      | CCE/SEI for Europe                                  | Corine Land Cover 2006 (22 classes)                 |

[revised manuscript text omitted]